# Bimodal fibrosis in a novel mouse model of bleomycin-induced usual interstitial pneumonia

Yoko Miura[1], Maggie Lam[2], Jane E Bourke[2] ⓘ, Satoshi Kanazawa[1] ⓘ

**Idiopathic pulmonary fibrosis is pathologically represented by usual interstitial pneumonia (UIP). Conventional bleomycin models used to study pathogenic mechanisms of pulmonary fibrosis display transient inflammation and fibrosis, so their relevance to UIP is limited. We developed a novel chronic induced-UIP (iUIP) model, inducing fibrosis in D1CC×D1BC transgenic mice by intra-tracheal instillation of bleomycin mixed with microbubbles followed by sonoporation (BMS). A bimodal fibrotic lung disease was observed over 14 wk, with an acute phase similar to nonspecific interstitial pneumonia (NSIP), followed by partial remission and a chronic fibrotic phase with honeycombing similar to UIP. In this secondary phase, we observed poor vascularization despite elevated PDGFRβ expression. γ2PF- and MMP7-positive epithelial cells, consistent with an invasive phenotype, were predominantly adjacent to fibrotic areas. Most invasive cells were *Scgb1a1* and/or *Krt5* positive. This iUIP mouse model displays key features of idiopathic pulmonary fibrosis and has identified potential mechanisms contributing to the onset of NSIP and progression to UIP. The model will provide a useful tool for the assessment of therapeutic interventions to oppose acute and chronic fibrosis.**

## Introduction

Interstitial pneumonia (IP) is a lung dysfunction with expanding fibrotic lesions due to the accumulation of ECM. Idiopathic pulmonary fibrosis (IPF) is one of the most serious chronic and progressive IPs, leading to death due to the loss of pulmonary function (Antoniou et al, 2014). Fibrotic foci in IPF are often covered by a cuboidal lining epithelium, attributed to squamous metaplasia of epithelial cells (Hogan et al, 2014; Hashisako & Fukuoka, 2015; Batra et al, 2018). The histopathology of IPF is defined by usual interstitial pneumonia (UIP), which is derived from various features such as honeycombing and bronchiolization of alveoli together with severe fibrosis (Hashisako & Fukuoka, 2015; Raghu et al, 2018). The UIP pattern is characterized by geographic heterogeneity, with patchy interstitial fibrosis alternating with normal parenchyma. In IPF, the UIP pattern is found mainly in subpleural, basolateral areas of the lungs, with scattered foci of fibroblasts in areas of dense acellular collagen (Smith et al, 2013). Most honeycomb cysts are surrounded by hyperplastic epithelium, and associated with traction bronchiolectasia (Hashisako & Fukuoka, 2015). IPF is also associated with the loss of endothelial cells and microvasculature, causing pulmonary hypertension (Nathan et al, 2007; Farkas et al, 2009). Nonspecific interstitial pneumonia (NSIP) and UIP are distinct histopathological patterns commonly observed in various interstitial pneumonias, with UIP being characteristic of IPF (Travis et al, 2008; Tabaj et al, 2015). The underlying mechanisms of these pathological changes in IPF are not yet fully defined.

The development of pulmonary fibrosis is most commonly studied in animal models using bleomycin, an oligopeptide antibiotic produced by *Streptomyces verticillus* (Hay et al, 1991; Organ et al, 2015). Although bleomycin has established chemotherapeutic effects against cancer, its side effects in patients include pathologic changes consistent with IPF (Adamson, 1976). In animal models, administration of single or multiple doses of bleomycin by either intra-tracheal (i.t.) instillation, osmotic pump, intravenous route, or intranasal delivery induces pulmonary fibrosis, results in significant dose-dependent mortality (Degryse et al, 2010; Mouratis & Aidinis, 2011; Moore et al, 2013; Jenkins et al, 2017). Bleomycin mediates DNA double-strand breaks (DSBs) and thus induces lung injury with severe inflammation (Mouratis & Aidinis, 2011). The half-life of bleomycin is from a few hours up to 21 h in the body. It triggers events seen in the onset of pulmonary fibrosis, namely, epithelial injury and alveolar infiltration of lymphoid cells such macrophages (Dorr, 1992; Cooper, 2000). In most animal studies, bleomycin-treated mice are analyzed 14–28 d after a single intra-tracheal dose. Acute inflammatory changes seen in C57BL/6 mice over this relatively short time-frame resemble those seen in lung diseases like nonspecific interstitial pneumonia (NSIP), but usually occur without the development of chronic symptoms unless higher or repeated doses of bleomycin are administered (Satoh et al, 2017; Tashiro et al, 2017). The efficacy of potential therapeutic agents is usually assessed at a time that overlaps with recovery from these short-term bleomycin models, and IP biomarkers such as

[1]Department of Neurodevelopmental Disorder Genetics, Nagoya City University Graduate School of Medical Sciences, Nagoya, Japan  [2]Department of Pharmacology, Biomedicine Discovery Institute, Monash University, Clayton, Australia

Correspondence: kanas@med.nagoya-cu.ac.jp

surfactant protein-D (SP-D) are not monitored (Murata et al, 2010). As such, it is often difficult to distinguish between the effects of treatment and spontaneous recovery (Moeller et al, 2008). Of note, current models of single bleomycin instillation in C57BL/6 mice do not generally result in the long-term histological characteristics seen with UIP in patients with IPF (Borzone et al, 2001; Redente et al, 2011; Moore et al, 2013; Limjunyawong et al, 2014; Tashiro et al, 2017).

Although the cause of IPF remains unknown, findings from studies of both clinical samples and short-term bleomycin models have implicated repetitive trauma of epithelial cells and dysregulation of repair processes mediated by TGF-$\beta$ in disease progression (Hashisako & Fukuoka, 2015; Jiang et al, 2020). Of particular note, blockage of TGF-$\beta$ signaling resulted in attenuation of fibrotic responses during wound healing in bleomycin-induced pulmonary fibrosis (Degryse et al, 2011), informing the current first-line treatment of IPF with pirfenidone and nintedanib (King et al, 2014; Flaherty et al, 2019).

To address these limitations, we have developed a new model of UIP, using D1CC$^{+/+}$×D1BC$^{+/+}$ transgenic mice (hereafter D1CC×D1BC tg), bred on a DBA/1J background. These mice were bred by crossing IP-susceptible D1CC and D1BC tg mice, which express class II transactivator (CIITA as an MHC class II transcriptional co-activator) and B7.1 (co-stimulatory signaling molecule), respectively (Fontes et al, 1999; Kanazawa et al, 2000; Miura et al, 2019). D1CC, D1BC, and D1CC×D1BC tg mice have chronic joint inflammation, as seen in patients with rheumatoid arthritis (RA) with additional interstitial lung disease (Kanazawa et al, 2006; Miura et al, 2019, 2021; Terasaki et al, 2019; Miura & Kanazawa, 2020). After induction of RA by low-dose type II collagen, wild-type DBA/1J mice did not develop pulmonary inflammation, whereas the tg mice had increased serum SP-D levels and chronic deposition of ECM in the lung and higher susceptibility to IP than in C57BL/6 mice (Schurgers et al, 2012; Terasaki et al, 2019; Miura et al, 2021). Histopathology of lung tissues in mice with joint inflammation was characterized as NSIP, defining this as a model of RA-interstitial lung disease.

Given this background, we hypothesized that bleomycin would induce features of NSIP and potentially UIP in D1CC×D1BC mice. To facilitate the development of a chronic model, we used a novel method to deliver a single dose of bleomycin (lower than usually administered in other models) mixed with microbubbles and sonoporation (hereafter called BMS), to minimize mortality while increasing drug delivery to the lung (Okada et al, 2005; Xenariou et al, 2010; Tashiro et al, 2017; Yamaguchi et al, 2017). This BMS mouse model induced bimodal fibrosis, alveolar bronchiolization and altered vascularity, mimicking key features of UIP in IPF, and enabled the characterization of specific phenotypes of the resident cells contributing to these pathological changes.

# Results

## BMS induces chronic IP

After a single i.t. instillation of BMS in D1CC×D1BC mouse (Fig 1A), disease progression was monitored by the IP biomarker, serum SP-D (Miura et al, 2019). The peak SP-D concentration after BMS was > 2,500 ng/ml at week 2, approximately threefold higher than after i.t. instillation of the same dose of bleomycin alone (Fig 1B). Serum SP-D levels declined markedly in both groups by 6 wk, with only the BMS group still slightly elevated above the cutoff value (>53.9 ng/ml) after 8 wk (Fig 1B inset). The selected dose was the mid-range of dose-ranging pilot studies using 0.96–1.6 mg/kg body weight of bleomycin (Fig S1 and Table S1). Sonoporation, either alone or in combination with microbubbles, did not induce sufficient trauma to cause the development of IP, as evidenced by its lack of effect on serum SP-D, whereas lower doses of BMS were also ineffective in the tg mice (Fig S2).

Body weight was decreased by 5~10% during the acute phase after BMS administration in most D1CC×D1BC mice, and slowly recovered until week 14 (Fig 1C). On the other hand, the total lung weight showed a bimodal pattern over 14 wk (Fig 1D). Total lung expression levels of SP-D were decreased when serum SP-D was at its peak during disease progression (Fig 1E). The patterns of serum and lung SP-D levels suggest that the maximal leakage from pulmonary parenchyma into the blood coincided with acute BMS-induced inflammation. Thus, there was incompatibility between serum SP-D and total SP-D in the whole lung during disease progression. We also examined total lung expression levels of the other C-type lectins, SP-A and -C, by Western blot (WB, representative images shown in Fig S3). SP-A levels were lower at week 2 than in untreated controls and tended to recover or be higher than control levels by week 14 in the chronic phase, whereas SP-C levels were slightly increased (Fig 1F and G). Finally, the mortality with this relatively low bleomycin dose was less than 20%, with only 2 deaths in a total of 14 mice over 14 wk (Fig 1H).

## BMS induces DNA damage lung restricted to AEC1, AEC2, and bronchiolar epithelial cells

We examined whether BMS-induced cell death during disease progression. Apoptotic or necrotic cells were observed in lung alveoli at 6–24 h (Fig 2A). The current findings are consistent with bleomycin-induced cell death occurring mainly within 1 d of administration Consistent with bleomycin-induced cell death occurring mainly within 1 d (Fig S4), there was no increase in cell death after day 3 (Hagimoto et al, 1997). The local effects of BMS seen on lung epithelial cells were not evident either in lung stromal cells or other tissues such as skin and liver (Fig S5).

Bleomycin is a radical transfer agent, which induces single- or double-strand DNA breaks in the nucleus. We examined the percentage of γH2AX-positive cells as a marker of this DNA damage at days 3, 7, and 14. Approximately 5% of total lung cells were γH2AX-positive at day 3, with numbers decreasing to similar levels as in untreated controls by week 2 (Fig 2B). Thus, bleomycin-induced DNA breaks may be limited to the first few days. Next, we investigated which cell types were susceptible to bleomycin. The vast majority of γH2AX-positive cells were podoplanin-positive AEC1s at day 3 (Fig 2C). The bronchiolar epithelium and AEC2 also displayed significant susceptibility to bleomycin, but fibroblasts and lymphocytes were rarely γH2AX positive (Fig 2D). Because 90–95% of the lung surface is covered with AEC1s, the bronchiolar epithelium and AEC2 may be relatively more sensitive to bleomycin-induced DNA breaks than AEC1, whereas fibroblasts may be resistant (McElroy & Kasper, 2004).

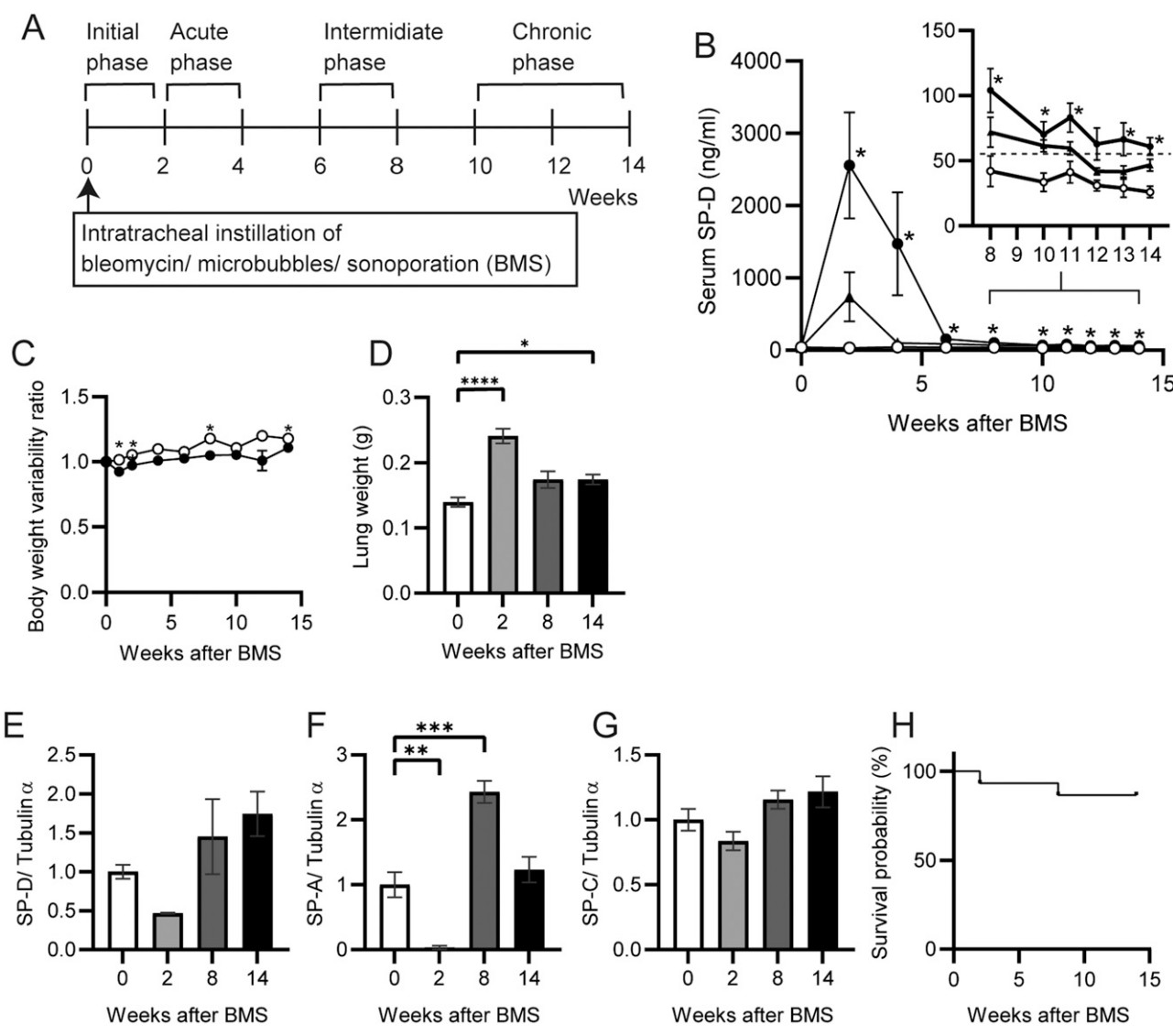

**Figure 1. Serum and lung SP-D levels during BMS-induced disease progression.**
**(A)** Timeline after i.t. instillation of bleomycin (1.28 mg/kg body weight) with microbubbles followed by sonoporation (BMS) in D1CC×D1BC mice, showing initiation, acute, intermediate, and chronic phases of IP progression. **(B)** Serum SP-D levels over 14 wk after BMS (closed circle), bleomycin alone (closed triangle), or vehicle (open circle) in D1CC×D1BC mice evaluated by ELISA. The dotted line in inset graph indicates serum SP-D cutoff value (53.9 ng/ml). Data are means ± SE of seven mice at each time point. Asterisks show *$P < 0.05$, compared with vehicle. **(C, D)** Body weight variability ratio and (D) total lung weight after BMS administration in D1CC×D1BC mice. Data are means ± SE from nine mice per group. Asterisks show *$P < 0.05$ and ****$P < 0.0001$. **(E, F, G)** SP-D (E), SP-A (F), and SP-C (G) in whole lung extracts at weeks 0, 2, 8, and 14 measured by WB. The relative signal intensity from densitometrical analysis was calculated using ImageJ Fiji, with tubulin-$\alpha$ used as a loading control. Data are means ± SE from three mice per group. Asterisks show **$P < 0.01$, and ***$P < 0.001$, compared with 0 wk. **(H)** Kaplan–Meier analysis of mouse survival post-BMS from weeks 0–14 of 14 mice.

## BMS induces bimodal fibrosis characterized by acute inflammation and chronic honeycombing

To measure fibrosis in BMS-induced D1CC×D1BC mice, we assessed Masson's Trichrome stained sections at selected time points over 14 wk (Figs 3A and S6). Analysis of digital images by ImageJ, Fiji, was performed to obtain fibrosis area ratio values (Fig 3B). Pulmonary fibrosis was evident by week 2, with peaks observed at week 2–6 in the acute phase and also at week 14 in the chronic phase. Some level of fibrosis was retained during remission with secondary diffuse fibrosis observed at the pleural surface at week 8 (Fig S6).

Notably, severe fibrosis with honeycomb structure, but not marked infiltration of lymphoid cells, was seen in the chronic phase. This is because the percentage of macrophages and T cells increased at week 2, but infiltrated cells were restored to control levels at weeks 8 and 14 (Fig S7).

The highest Ashcroft score for lung fibrosis was associated with the honeycomb structure seen in the specimens at week 14 (Fig 3C). A similar bimodal pattern of fibrosis was observed using HistoIndex stain–free imaging to detect collagen, combined with FibroIndex analysis to determine collagen area and collagen fiber density (Fig 3D–F). The total amount of fibrillar collagen, measured by a Sircol

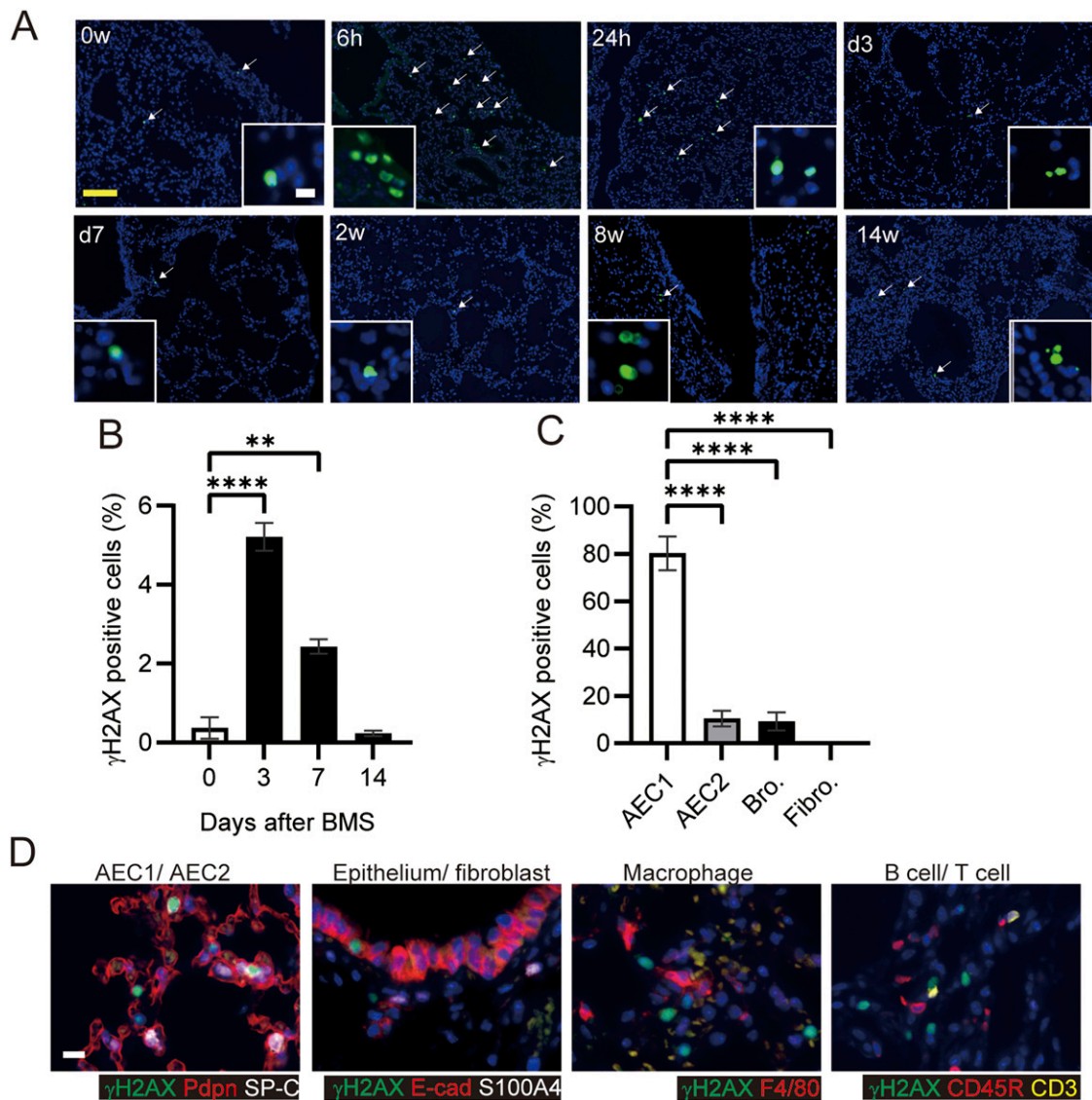

**Figure 2. DNA damage in AEC1, AEC2, and bronchioles, but not in fibroblasts.**
**(A)** TUNEL assays in mouse lungs at 0, 6, and 24 h, 3 and 7 days, and 2, 8, and 14 weeks after BMS administration. Arrows indicate cell death. **(B)** Percentage of γH2AX-positive cells at days 0, 3, 7 and 14 in mice treated with BMS. Images were captured and analyzed from five randomly selected fields on each slide at a magnification of 200×. Data are means ± SE from five mice per group. Asterisks show **$P < 0.01$, ****$P < 0.0001$, compared with 0 wk. **(C)** Percentage of γH2AX-positive cells in AEC1 (podoplanin), AEC2 (SP-C), bronchiolar epithelium (E-cadherin, Ecad), and fibroblasts (S100A4) at day 3. Images were captured and analyzed from 10 randomly selected fields on each slide at a magnification of 200×. Data are means ± SE from three mice per group. Asterisks show ****$P < 0.0001$, compared with the percentage of γH2AX-positive cell in different cell type. **(D)** Immunohistochemical staining of γH2AX (green), podoplanin (Pdpn, red), SP-C (white), E-cadherin (E-cad, red), S100A4 (white), F4/80 (red), CD45R (red), and CD3 (yellow). Scale bar = 100 $\mu$m (yellow) and 10 $\mu$m (white).

assay for soluble collagen, was also increased at weeks 2, 8, and 14 (Fig 3G). By comparison, only weak IP was observed in D1CC alone, D1BC alone, and their genetic background DBA/1J mice (Ashcroft scores in all groups <3 at 14 wk), suggesting minimal susceptibility for fibrosis in response to BMS administration in DBA/1J mouse (Fig S8).

Expression of collagen type I (Col1), measured by WB, showed a similar pattern as seen for the other measures of fibrosis (Fig 3H). Immunohistochemical staining and WB confirmed that the increased αSMA was localized to fibrotic areas evident after BMS administration (Fig S9). We performed double-immunohistochemical staining for Col1 and S100A4 (also called fibroblast specific protein 1). The expression of both proteins overlapped at week 2 and 8 in the fibrotic area; however,

less S100A4 expression was observed in the Col1-rich fibrotic area at week 14 (Fig 3I). Expression of TGF-$\beta$, a robust stimulator of fibrosis that also inhibits cell proliferation, was measured in the whole lung by WB. Although the changes were not significant, TGF-$\beta$ tended to increase at week 14 only, and not at weeks 2 and 8 (Fig 3J). Overall, our analysis established a highly consistent pattern of bimodal fibrosis in the BMS model.

### Extension of fibroblasts in fibrotic foci in chronic phase

Further characterization of fibrotic changes in BMS-induced D1CC×D1BC mice was performed by WB of whole lung. Vimentin (fibroblast marker) and PDGFR$\beta$ (pericytes) were significantly increased at week 2, and

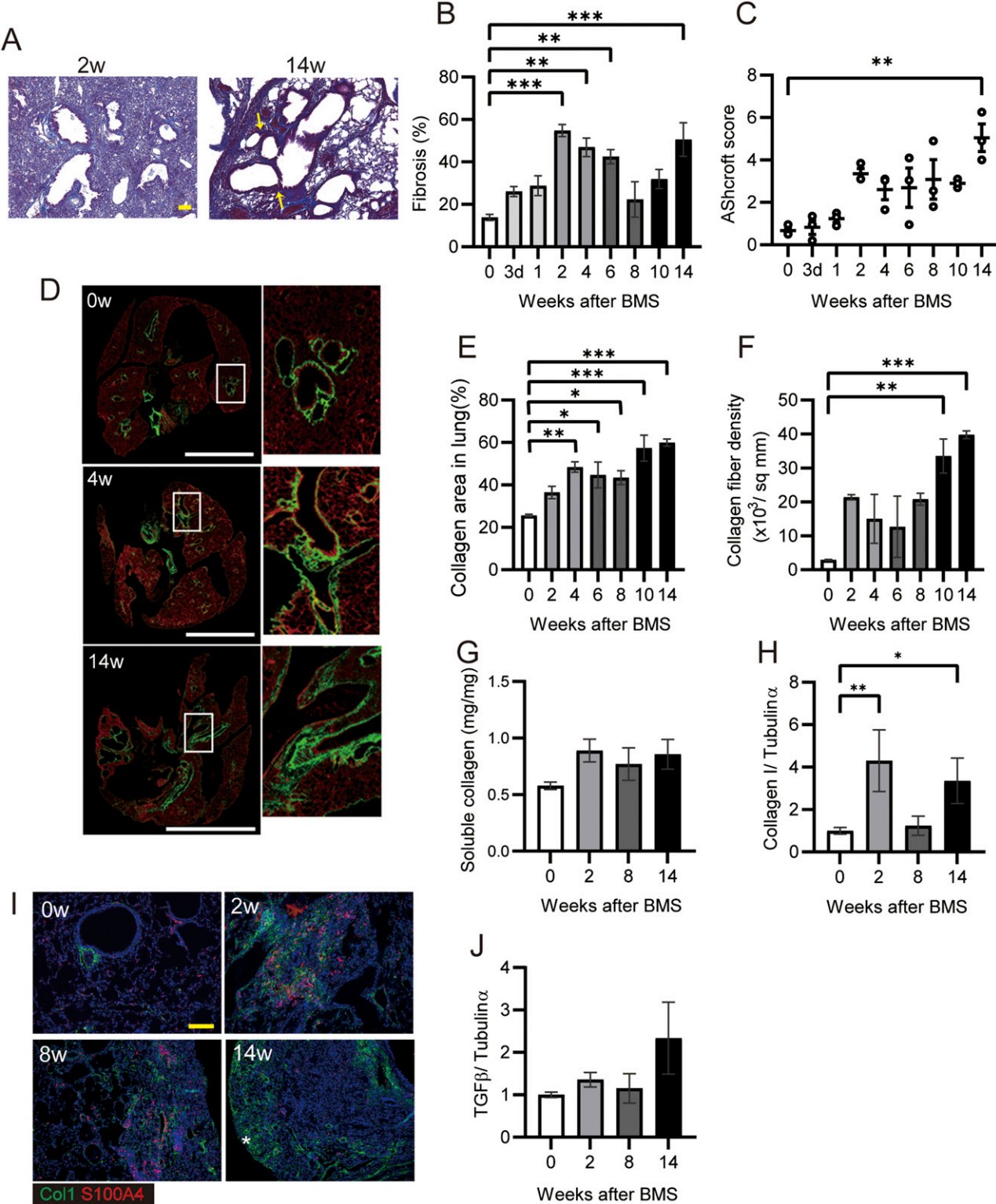

**Figure 3.   Bimodal fibrotic stages in IP progression.**
**(A)** Representative histopathological micrographs of Masson's trichrome–stained sections from BMS-induced D1CC×D1BC mice. In acute phase at week 2, many infiltrated cells were observed. In the chronic phase at week 14, honeycombing was indicated by yellow arrows. **(B)** Quantitative analysis of fibrosis from Masson's trichrome stained sections as ratios of ECM area (blue) to total lung area, as calculated by ImageJ, Fiji. n = 6–7 per each group. **(C)** Quantitative analysis of Ashcroft score (averaged from scores by two individuals, Y Miura and S Kanazawa, performed in a blinded manner). n = 3 per group. **(D)** Representative images at weeks 0, 4, and 14 captured with HistoIndex show the fibrotic area (mostly collagen I and III, green) and the entire lung area (cell structure, red). **(E, F)** Collagen deposition and (F) collagen

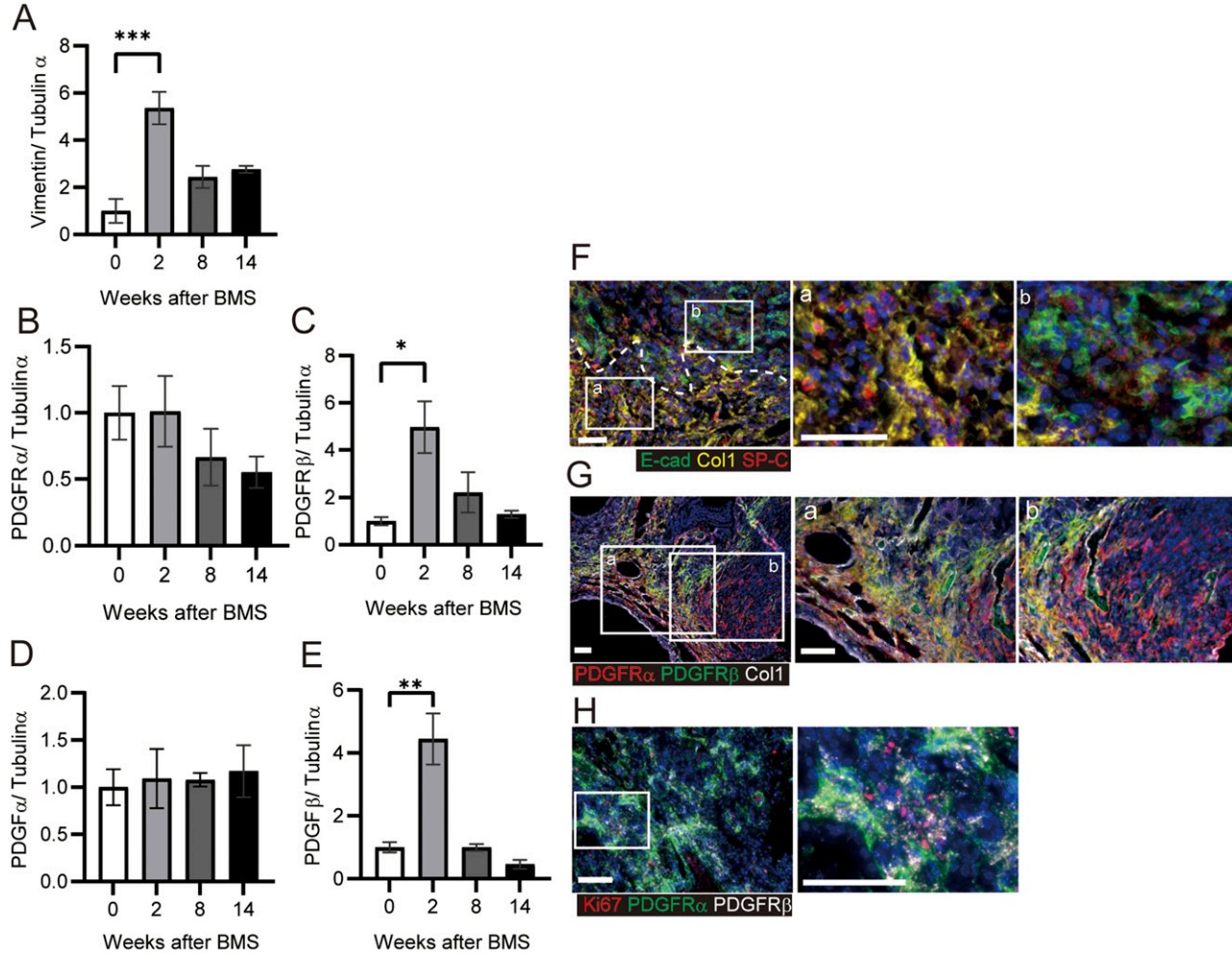

**Figure 4. Extension of fibroblasts rather than proliferation in fibrotic foci in chronic phase.**
**(A, B, C, D, E)** Expression of (A) vimentin, (B) PDGFRα, (C) PDGFRβ, (D) PDGFα, and (E) PDGFβ at weeks 0, 2, 8, and 14 determined by WB and ImageJ, Fiji. Data are presented means ± SE of three mice for each week. Asterisks shows *$P < 0.05$, **$P < 0.01$, and ***$P < 0.001$, compared with 0 wk. **(F, G, H)** Representative immunostaining at week 14 showing (F) fibrotic area (Col1, yellow), bronchiolar epithelial cells (E-cadherin (E-cad), green), and AEC2s and bronchiolar epithelial cells (SP-C, red), (G) fibrotic area (Col1, white), PDGFRα (red) and PDGFRβ (green), and (H) Ki67 (red), PDGFRα (green), and PDGFRβ (white). Scale bars = 50 μm. All specimens for immunostaining were obtained at week 14.

decreased at later time points, whereas expression of PDGFRα (fibroblasts) only showed a gradual decline (Fig 4A–C). Next, we analyzed the expression of PDGFR ligands, PDGFα and PDGFβ, in the whole lung. Only PDGFβ was elevated at week 2, but this was not maintained during disease progression (Fig 4D and E). This suggests that PDGFβ-mediated growth of fibroblasts may be limited to the acute phase.

To further focus on fibrotic foci at week 14, we performed multiple immunohistochemical analyses using antibodies against E-cadherin (epithelial cell marker), SP-C (epithelial cells and AEC2s), and Col1 (fibroblasts). A Col1⁺ fibrotic area was localized adjacent to, but distinct from, another area consisting of E-cadherin⁺/SP-C⁺ epithelial cells (Fig 4F). Further analysis in this Col1-positive region showed that PDGFRα and PDGFRβ double-positive areas were predominant over PDGFRα single-positive areas (Fig 4G). Most Ki67-positive cells were excluded from the area expressing PDGFRs (Fig 4H). These data suggest that in the absence of increased PDGF at week 14, most fibroblasts tend to aggregate or extend rather than continue to proliferate in the fibrotic region.

fiber density in 2-μm-thick unstained specimens captured by HistoIndex and analyzed using FibroIndex software. n = 3 per each group. **(G)** Quantitative analysis of total soluble collagen using sirius red. n = 5 per each group. **(H)** Collagen I expression determined by WB and ImageJ, Fiji. n = 3 per each group. **(I)** Immunohistochemical staining of Col1 (green) and S100A4 (red) at weeks 0, 2, 8, and 14. Asterisks indicates subpleural hyperplasia. **(J)** TGF-β expression determined by WB and ImageJ, Fiji. **(A, D, I)** Scale bars are 100 μm in (A) and (I), and 5 mm in (D). **(G, H, J)** n = 3 per each group. Data are presented means ± SE. Asterisks show *$P < 0.05$, **$P < 0.01$, and ***$P < 0.001$, compared with week 0.

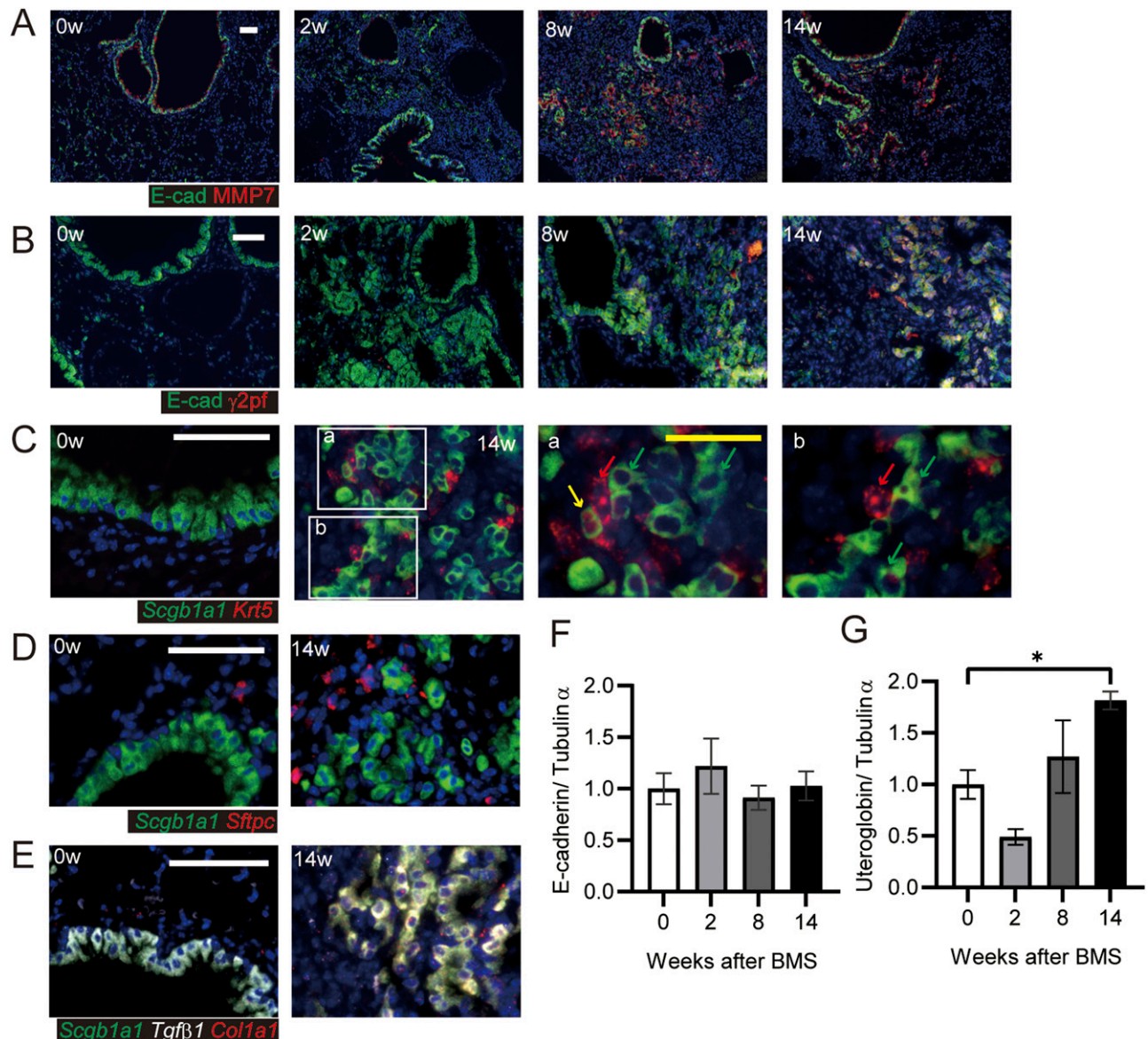

**Figure 5. Invasive bronchiolar epithelial cells in chronic phase.**
**(A, B)** Immunostaining for invasive bronchiolar epithelial cells using (A) E-cadherin (E-cad, green) and MMP7 (red) or (B) E-cadherin (E-cad, green) and N-terminal proteolytic fragments of Laminin, γ2pf (red). **(C, D, E)** In situ hybridization for invasive bronchiolar epithelial cells at weeks 0 and 14. **(C, D, E)** *Scgb1a1* (green) and *Krt5* (red), (D) *Scgb1a1* (green) and *Sftpc* (red), and (E) *Scgb1a1* (green), *Tgfβ1* (white), and *Col1a1* (red). Each arrow indicates *Krt5* (red), *Scgb1a1* (green), and *Krt5* and *Scgb1a1* (yellow)-positive cells in C a and b. Scale bars = 50 μm (white) and 25 μm (yellow). **(F, G)** E-cadherin and (G) uteroglobin expression determined by WB and ImageJ, Fiji. Data are presented means ± SE of three mice. Asterisks show *P < 0.05, compared with week 0.

## Hyperplastic bronchiolar cells with invasive phenotype in the damaged lung

Hyperplastic epithelial cells were also prominent and widely distributed after BMS induction (Fig 5A). Co-staining of these E-cadherin–expressing cells was performed for matrix metalloproteinase-7 (MMP7) and the N-terminal proteolytic fragments of laminin γ2 chain (γ2pf) to assess how these cells might migrate or expand in the damaged lung (Miyazaki et al, 2016). Although epithelial cells in the acute phase did not express these markers, MMP7 and γ2pf co-expression with E-cadherin was clearly evident after the

intermediate phase (Fig 5A and B). This was associated with destruction of the basement membrane and cell migration into the alveolar space.

These invasive cells showed positive staining for secretoglobin, family 1A1 (*Scgb1a1*, which encodes uteroglobulin) and/or keratin 5 (*Krt5*), markers of club and basal cells, respectively (Fig 5C) (Hogan et al, 2014). *Scgb1a1*-positive invasive cells also expressed the fibrotic markers transforming growth factor-β1 (*Tgf-β1*) and collagen1A1 (*Col1a1*), suggesting the spread of fibrosis into areas of epithelial hyperplasia. These cells differed from bronchioalveolar stem cells (BASCs) because of this more fibrotic phenotype and

less *Sftpc* expression (Fig 5D and E) (Kim et al, 2005; Jiang et al, 2020).

We next performed WB for total E-cadherin and uteroglobin expression in the lung. Uteroglobin expression, but not E-cadherin was increased at week 14 (Fig 5F and G). This suggested that the number of invasive epithelial cells was increased, whereas the cell-to-cell contact of bronchiolar epithelial cells was not increased. Next, we tested whether these cells had undergone epithelial mesenchymal transition (EMT) by performing immunohistochemistry for the fibroblast marker S100A4. Evidence of E-cadherin and S100A4 co-staining was apparent at week 14 only (Fig S10). Taken together, these data suggest migratory epithelial cells lose their polarity and become dysplastic with or without EMT.

### Poor reconstitution of blood vessels because of lack of CD31-positive endothelial cells

Poor vascularization results in pulmonary hypertension in patients with IPF (Nadrous et al, 2005). We analyzed the potential reconstitution of blood vessels after destruction of the lung structure after BMS administration. Initially, we investigated the localization of VEGF and PDGFRβ as key proteins related to vascularization. There was no appropriate blood vessel formation after BMS administration (Fig 6A). VEGF expression was evident in both normal as well as invasive bronchiolar epithelial cells (Fig 6A and B) (Fehrenbach et al, 1999). In the fibrotic regions at week 14, PDGFRβ-positive fibroblasts aggregated in regions distinct from areas of VEGF-positive staining, whereas CD31 expression was low or almost undetectable (Fig 6A) (Renzoni et al, 2003). Whole lung expression of CD31 and VEGF was also examined by WB, along with pigment epithelial derived growth factor (PEDF), which opposes angiogenesis. Acute BMS-induced decreases in VEGF and CD31 expression were not restored in all phases after BMS administration (Fig 6C and D). A transient increase in PEDF expression at week 2 returned to near control levels at weeks 8 and 14 (Fig 6E).

Further histopathological features were evident in the hyperplastic interstitial area. Decreased numbers of AEC1 and AEC2 were seen adjacent to hyperplastic epithelial cells (area a in Fig 6F); Blood vessels and endothelial cells remained in the interstitial area, but CD31 expression was low (area b in Fig 6F). Poor microvascular structure in the chronic phase was confirmed from micro-CT images of the lung vasculature filled ex vivo with Microfil. At 14 wk post-BMS, there were fewer subpleural capillaries, and decreased volume of blood vessels (Fig 6G and H). Overall, the marked loss of endothelial cells in the lung may result in poor reconstitution of blood vessels, despite the aggregation of PDGFRβ-positive cells that may have pericyte functions in the fibrotic area.

## Discussion

A novel iUIP model using BMS as a stimulus was characterized by multi-step disease progression, most notably bimodal fibrosis with honeycombing. We propose that the application of this BMS method has increased the short half-life of bleomycin and/or provided a higher localized dose to induce these changes, but with lower mortality relative to previous studies. In the current study, a metaplastic respiratory epithelium that expressed E-cadherin was observed after administration of BMS, similar to that seen in IPF. The chronological features of multiple markers of disease progression in this iUIP mouse model induced by BMS are summarized in Fig 7. The clear pattern of bimodal fibrosis observed in the acute and chronic phases was similar to the histopathology of NSIP and UIP, respectively.

Bleomycin-induced DNA damage in the acute phase was limited to epithelial cells, with no overt DSBs detected in fibroblasts. This suggests that the secondary fibrosis with honeycombing may be indirectly regulated by these damaged epithelial cells in addition to fibroblasts. This would implicate NSIP as a previous stage of UIP, a possibility that could not be previously explored in conventional models.

During the acute phase, bronchiolar epithelial cells expressing a putative EMT biomarker MMP7 rather than the migratory marker S100A4 extended into the lung interstitium (Rosas et al, 2008; Kropski et al, 2015; White et al, 2016). Enhanced migration from the area of hyperplastic bronchiolar epithelia was consistent with elevated MMP7 expression leading to destruction of the basement protein laminin. Expression of γ2pf, an additional invasive marker commonly seen in cancer cells, was increased in MMP7-positive cells in the intermediate and chronic phases. Thus, loss of cell-to-cell contact may cause hyperplastic alveoli to migrate into the lung interstitium to increase bronchiolization during the progression of secondary fibrosis. Of note, expression of MMP7 was elevated along with spontaneous lung fibrosis in a *Sftpc* mutant expressing mouse model (Nureki et al, 2018).

*Krt5* expression was relatively weaker by week 14 in *Scgb1a1*-positive invasive cells, suggesting that they may be similar to lineage-negative epithelial stem/progenitor cells (LNEPs) or multipotent progenitor cells (Hogan et al, 2014; Vaughan et al, 2015). Although this finding suggested that the cells had become dysplastic, they were different from squamous cell carcinoma in this stage. In a single cell analysis of epithelial cells in patients with IPF, the predominant cells were progenitor subpopulations, which express *Scgb1a1*, *Krt5*, *Krt8*, and *Trp63* (including ΔNp63), and *TGF-β* as shown in the current model. The presence of these invasive epithelial cells may lead to failure to regenerate the lung epithelium during chronic fibrosis (Vaughan et al, 2015; Jiang et al, 2020). Further characterization of invasive epithelial cells in the intermediate and the chronic phases is required to implicate NSIP and DNA damage in epithelial cells as a requirement for the continuous pathological changes that contribute to chronic disease progression in some patients with IPF. Isolation of bronchiolar epithelia in specific areas of the lung using lineage tracing techniques may reveal patterns of gene expression to provide insights into potential factors that promote fibrosis and EMT.

Poor vascularization persisted from acute to chronic phases in BMS-induced D1CC×D1BC tg mice. The expression of pro-angiogenic VEGF was decreased at acute phase, whereas the corresponding inhibitory factor PEDF was transiently increased. It remains difficult to estimate the amount of active rather than total TGF-β, which is also a putative regulator for PEDF. Acute inflammation by bleomycin could affect the activation of TGF-β directly and/or indirectly

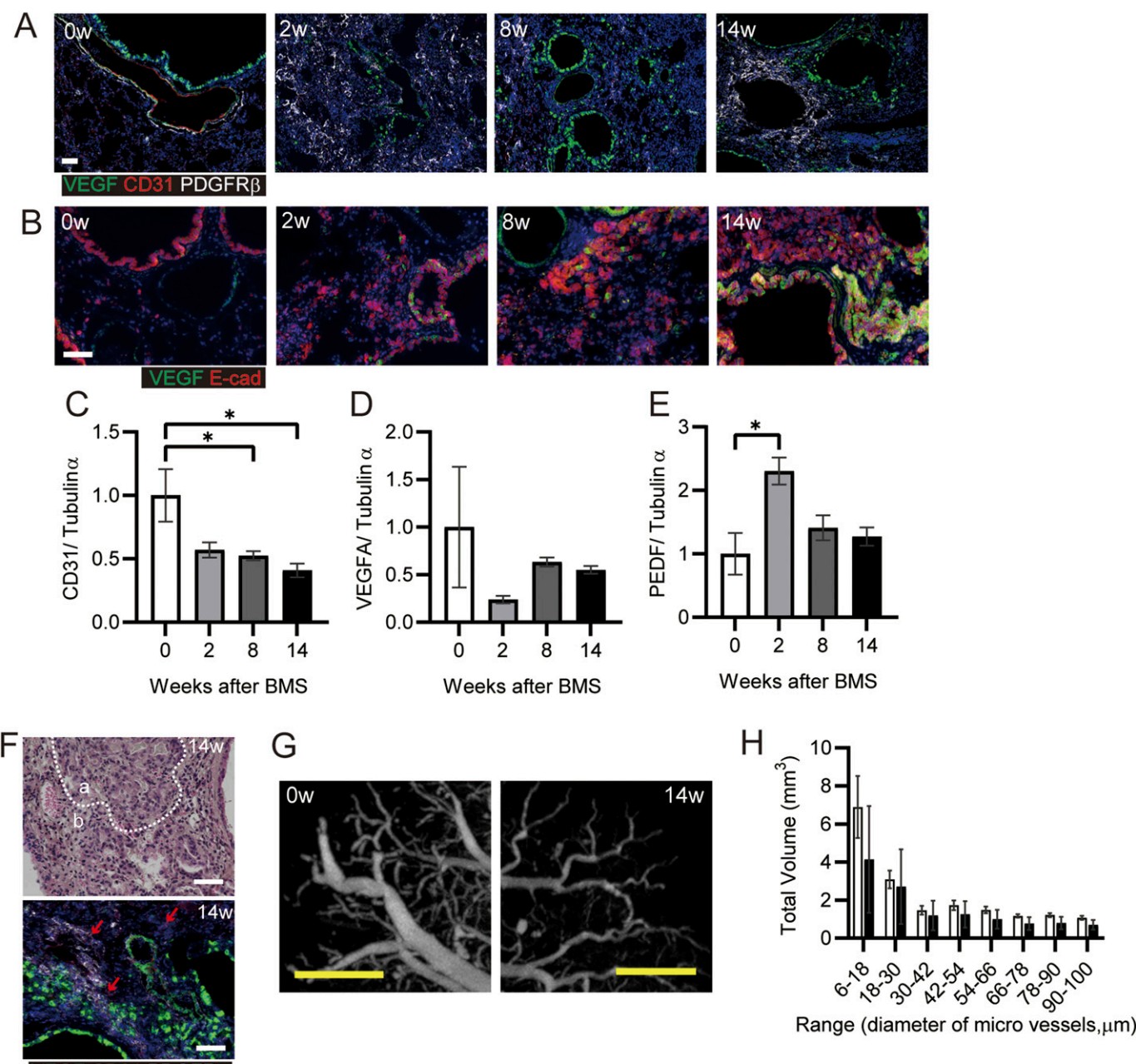

**Figure 6. Poor reconstitution of blood vessels in chronic phase.**
**(A, B)** Immunohistochemical staining for VEGF (green), CD31 (red), and PDGFRβ (white) or (B) VEGF (green) and E-cadherin (Ecad, red) at weeks 0, 2, 8, and 14. **(C, D, E)** CD31, (D) VEGFA, and (E) PEDF expression was determined by WB and ImageJ, Fiji. Data are presented means ± SE of three mice for each week, 0, 2, 8 and 14. Asterisk shows *$P < 0.05$, compared with week 0. **(F)** Representative hematoxylin eosin staining (upper) and immunohistochemical staining for VEGF (green), CD31 (red), and PDGFRβ (white) (lower). "a" indicates hyperplastic area with bronchiolar epithelial cells, whereas "b" indicates hyperplastic interstitial area at week 14. Arrows indicate CD31-positive cells. **(G)** Representative subpleural images at weeks 0 and 14 showing blood vessels, visualized using Microfil and micro-CT. **(H)** The total volume was calculated for each range of microvessel diameter in the whole lung after ex vivo Microfil application. Data are presented as means ± SE of three mice for each week, 0 and 14. **(A, B, F, G)** Scale bars = 50 μm (white), 1 mm (yellow).

through proteinase activation, resulting in fibrosis and vascular destruction. Although there was a persistent decrease in expression of CD31 simultaneously with VEGF during acute phase, this molecule may be more important for the transmigration of lymphocytes such as neutrophils, rather than for vascularization (Dangerfield et al, 2002). These combined findings suggest a complex interplay between the epithelium, fibroblasts and endothelium in the initiation and progression of fibrosis. However, whether these observations are common phenomenon induced by bleomycin treatment or whether this poor vascularization persists only in BMS-induced D1CC×D1BC mice requires further investigation.

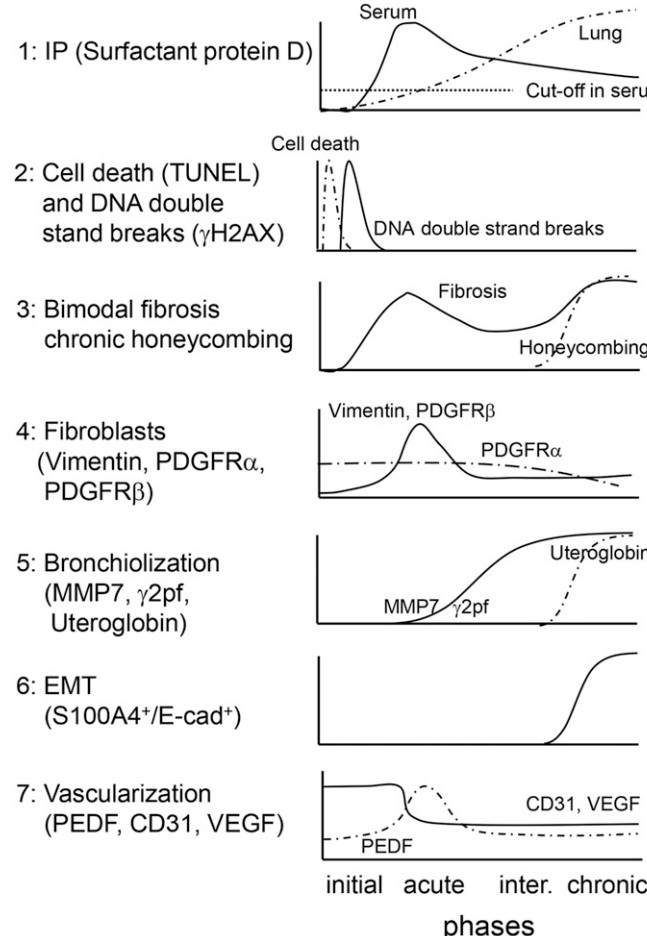

**Figure 7. Chronological features in iUIP mouse model.**
(1) Serum SP-D levels immediately increased in the acute phase, whereas total lung SP-D increased towards the chronic phase rather than the acute phase. (2) Cell death and double-strand breaks in alveolar epithelial cells were induced by BMS administration. (3) Bimodal fibrosis was found in the acute and the chronic phases. Honeycombing was observed from the intermediate to chronic phases. (4) Fibroblast expansion rather than growth was evident in the acute phase. (5) Invasive epithelial cells, which expressed MMP7 and γ2pf, were observed in the intermediate and the chronic phases. (6) Some invasive epithelial cells expressed both E-cadherin and S100A4, suggesting that epithelial mesenchymal transition occurring in the intermediate and chronic phases. (7) Poor vascularization was evidenced by reduction of CD31 and VEGF expression throughout the entire period after BMS administration. PEDF expression was up-regulated only in the acute phase.

In summary, we have demonstrated that D1CC×D1BC mice develop acute IP, bimodal fibrosis, and chronic honeycombing in response to BMS. Critically, chronic IP and fibrosis were notably absent when intra-tracheal bleomycin alone was given to D1CC×D1BC mice or when BMS was administered to DBA/1J mice, demonstrating the importance of both the method of administration and the transgene in this model. It will be of interest to determine whether other bleomycin-susceptible mouse strains, such as the more commonly used C57BL/6 mice, have a similar amplification of the response to a single intra-tracheal dose of bleomycin when administered as BMS, as this could support the widespread use of this novel method of administration. Although it

remains clear that the mechanisms underlying fibrosis are complex, our iUIP model will facilitate further research on chronic IP and potential targets for new drug therapy.

## Materials and Methods

### Mice and BMS administration

D1CC×D1BC tg mice, bred on a DBA/1J background, were housed in a pathogen-free animal care facility of Nagoya City University Medical School in accordance with institutional guidelines (Kanazawa et al, 2006; Miura et al, 2019). Mice were anesthetized with isoflurane and the chest hair was shaved for sonoporation. Bleomycin (0.512 mg/ml in normal saline, Nippon Kayaku) was mixed with an equal amount of microbubbles (Ultrasound Contrast Agent SV-25; NepaGene) and administered via the i.t. route by a spray nebulizer (40 µl/mouse, 1.28 mg/kg body weight; Natsume) before sonoporation on the chest by 1.0 W/cm² for 1 min (Sonitron GTS Sonoporation System; NepaGene). Induction of IP was monitored by measurement of serum SP-D levels. Preliminary tests confirmed that bleomycin has a uniform distribution in the lung. Under these conditions, there was no tendency to bias the sampling for assessment of lesion sites.

### ELISA

Serum was collected from the external jugular vein of each mouse from week 0–14 after BMS administration, aliquoted and stored at −80°C. Quantitation of serum SP-D concentrations was determined using ELISA according to the manufacturer's instructions (Rat/Mouse SP-D kit; Yamasa). The serum concentration of SP-D was measured in more than 20 BMS-untreated D1CC×D1BC mice, and the mean serum concentration of SP-D plus two times the SD was used as the cutoff value of SP-D, which was 53.9 ng/ml.

### Western blot

Lung samples were homogenized in RIPA buffer, containing 20 mM Tris–HCl, pH 7.4, 150 mM NaCl, 1% Triton-X100, 0.5% deoxycholate sodium, 0.1% sodium dodecyl sulfate, 1 mM EDTA, 10 mM BGP, 10 mM NaF, 1 mM Na₃VO₄, and protease inhibitor cocktail. The extracts were sonicated for 10 min and centrifuged at 13,000$g$ for 15 min. The supernatants were separated on 8, 14, or 18% SDS–PAGE and transferred onto polyvinylidene difluoride membranes. Protein-transferred membranes were blocked in 5% nonfat milk/TBS-T buffer at room temperature for 60 min. For Western blot, the following primary antibodies were used: rabbit anti-COL1A1 (Boster Biological Technology), rabbit anti-αSMA, rabbit anti-vimentin, rabbit anti-PDGFRα, rabbit anti-PDGFRβ and rabbit anti-E-cadherin (Cell Signaling Technology), goat anti-SP-D, mouse anti-TGF-β and goat anti-Serpin F1/PEDF (R&D Systems), rabbit anti-Uteroglobin, rabbit anti-CD31 and rabbit anti-VEGFA (Abcam), rabbit anti-SP-A, mouse anti-PDGFA and mouse anti–Tubulin-α (Santa Cruz), rabbit anti-SP-C (Hycult Biotech), and rabbit anti-PDGFB (Bioss Antibodies) antibodies. ECL anti-rabbit, anti-mouse, or

anti-goat IgG, horseradish peroxidase linked antibodies were used as a secondary antibody (GE Healthcare). Goat anti-mouse IgM-HRP was used as a secondary antibody for mouse anti–Tubulin-α (Santa Cruz). Each signal was detected using Immunostar Zeta or LD (Fuji film) and the Amersham Imager 600 series (GE Healthcare). Densitometry values used for statistical analysis of the expression levels of each protein were evaluated by ImageJ, Fiji.

### Analysis of lung sections for cellular and pathological markers

Lungs were collected at 0, 6, 12, and 24 h; 3 and 7 days; and 2, 4, 6, 8, 10, and 14 weeks after BMS administration and fixed overnight in 4% paraformaldehyde diluted in PBS and then embedded in paraffin before 2-μm-thick sections were cut. De-paraffinized sections were stained with hematoxylin–eosin (H&E) or Masson's trichrome to assess inflammation and fibrosis or subjected to conventional TUNEL assay according to the manufacturer's instructions to detect apoptotic cells (Mebstain Apoptosis Tunel kit, MBL).

For immunohistochemistry, the de-paraffinized sections were stained with the following primary antibodies: rabbit anti-E-cadherin, rabbit PDGFRα, rabbit PDGFRβ, rabbit anti-αSMA and rabbit anti-MMP7 (Cell Signaling Technology), mouse anti-phospho-H2AX and rabbit anti-S100A4 (Merck Millipore), rabbit anti-CD31 and rabbit anti-VEGFA (Abcam), rabbit anti-SP-C (Hycult Biotech), rat anti-Podoplanin and rabbit anti-PAD4 (MBL), rat anti-F4/80 (Bio-Rad), rabbit anti-CD3 (Genemed Biotechnologies), rat anti-PTPRC/CD45R (Aviva Systems Biology), rabbit anti-collagen I (Novus Biologicals), rat anti-Ki67 (Dako), and mouse anti-γ2pf (Funakoshi). Histofine Simple Stain Mouse MAX-PO secondary antibodies were used and visualized using the Histofine SAB-PO (M) kit (Nichirei Biosciences). For immunofluorescence, the Opal multiplex fluorescent immunohistochemistry system (Akoya Biosciences) were used according to the manufacturer's protocol. All images were captured and assessed for statistical analyses using BZ-X analyzer (Keyence) and ImageJ, Fiji (Schindelin et al, 2012).

### Quantitative analysis of phospho-H2AX (γH2AX)-positive cells

Immunohistochemical staining of γH2AX was performed to assess the percentage of DNA-damaged cells at day 0, 3, 7, and 14 after BMS administration, using hematoxylin as a counter stain. Five images (at a magnification of ×200) from each lung section were captured randomly and the percentage of γH2AX-positive cells was calculated by ImageJ, Fiji. For calculation of the percentage of each lung cell type within the γH2AX-positive cells, multi-color immunohistochemistry was performed for γH2AX, podoplanin (alveolar epithelial cell, AEC1), SP-C (AEC2), E-cadherin (bronchial epithelial cell), and S100A4 (fibroblast). Ten images (at a magnification of ×200) from each lung section were captured randomly and analyzed by hybrid cell count (Keyence).

## Ashcroft Score

The severity of lung fibrosis and destruction of lung structure including formation of honeycomb structure were evaluated by Ashcroft score using scanned Masson's trichrome–stained sections (Ashcroft et al, 1988). For analysis, 10 images from ×100 micrograph of whole lung sections were randomly selected from three mice for each day and week. These randomly selected images were individually assessed for IP severity by Y Miura and S Kanazawa in a blinded manner with an index of 0–8: 0, normal lung; 1, minimal fibrous thickening of alveolar or bronchiolar walls; 2–3, moderate thickening of walls without obvious damage to lung architecture; 4–5, increased fibrosis with definite damage to lung structure and formation of fibrous bands or small fibrous masses; 6–7, severe distortion of structure and large fibrous areas, evidence of "honeycomb lung"; 8, total fibrous obliteration of the field (Hubner et al, 2008).

### Fibrosis ratio

Images showing the area of fibrosis represented in blue (ECM-deposition) by Masson's trichrome staining (20 min for aniline blue staining) were captured by BZ-X analyzer (Keyence) and analyzed by ImageJ, Fiji. Data were calculated as fibrosis ratio, with ECM area divided by total lung area.

### HistoIndex

As an additional novel measure of lung fibrosis, the Genesis 200 (HistoIndex, located in Pharmacology, Monash University) was used to visualize unstained lung sections using Second Harmonic Generation to detect collagen as green light emission, with Two-Photon Excitation Fluorescence-based microscopy showing lung tissue in red. Images were captured with laser settings adjusted to 0.65 at 20× magnification with 512 × 512 pixels resolution and then assessed by the FibroIndex application for morphological analysis of interstitial collagen area and fiber density within the tissue (Goh et al, 2019).

### In situ hybridization

In situ hybridization for *Scgb1a1*, *Sftpc*, *Krt5*, *Col1a1*, and *Tgf-β1* was performed using the RNAscope Multiplex Fluorescent Reagent Kit v2 (Advanced Cell Diagnostics), according to the manufacturer's instructions.

### Soluble collagen assay

Soluble collagen content from whole lung extracts was determined by the Sircol assay (Biocolor), according to the manufacturer's instructions. Sircol dye bound to collagen was evaluated by a microplate reader at 555 nm.

### Micro-computed tomography

Mice were anesthetized with isoflurane and the abdominal and chest cavities were opened. The lung vasculature was perfused with PBS via the right ventricle and then fixed using 4% paraformaldehyde in PBS. Microfil (Flow Tech Inc.) was then injected via the same route and incubated for 1 h at room temperature (Phillips et al, 2017). Whole lung was harvested and fixed again overnight in

4% paraformaldehyde in PBS. All lungs were stored in 70% ethanol at 4°C until micro-CT scanning. All micro-CT images were acquired using the whole lung with a SkyScan 1276 (Bruker). Each volume was calculated for micro vessels with a diameter of less than 100 $\mu$m.

## Statistical analyses

The mortality rate in the iUIP mouse model was calculated by the Kaplan–Meier method (Prism9, GraphPad). Differences between non-instillation (0w) and the other groups were evaluated by one-way ANOVA followed by Dunnett's test for parametric data and Dunn's test for nonparametric data. Tukey's post hoc test was used for multiple comparisons of the percentage of $\gamma$H2AX-positive cells in different cell types. Values of $P < 0.05$ were considered statistically significant.

# Supplementary Information

# Acknowledgements

We thank Drs M Murata and T Numano for their help in all aspects of this work. We acknowledge the assistance of Dr Andre Tan from HistoIndex for training in Genesis 200 imaging and FibroIndex analysis. Grant support: This work was supported by grants-in aid from the Ministry of Education, Culture, Sports, Science and Technology (MEXT)/JSPS KAKENHI Grant Number JP 26461470, 23591444, 17K09982, and 17K16055. Grant-in-Aid for Research in Nagoya City University Grant Number 1943005, personal donation by T Furuya, and a Project Grant from the National Health and Medical Research Council Australia Grant Number 1165690.

## Author Contributions

Y Miura: formal analysis, validation, visualization, methodology, and writing—original draft, review, and editing.
M Lam: data curation, visualization, and writing—original draft, review, and editing.
JE Bourke: data curation, visualization, and writing—original draft, review, and editing.
S Kanazawa: conceptualization, resources, data curation, software, formal analysis, supervision, funding acquisition, validation, investigation, visualization, methodology, project administration, and writing—original draft, review, and editing.

## Conflict of Interest Statement

The authors declare that they have no conflict of interest.

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
