## [Reviewer comments · Life Science Alliance]

Life Science Alliance

Bimodal fibrosis in a novel mouse model of bleomycin-induced usual interstitial pneumonia

Satoshi Kanazawa, Yoko Miura, Maggie Lam, and Jane Bourke

DOI: <https://doi.org/10.26508/lsa.202101059>

Corresponding author(s): Satoshi Kanazawa, Nagoya City University Graduate School of Medical Sciences

Review Timeline:

Submission Date:	2021-02-17
Editorial Decision:	2021-07-06
Revision Received:	2021-09-26
Editorial Decision:	2021-10-13
Revision Received:	2021-10-20
Accepted:	2021-10-21

Transaction Report:

July 6, 2021

Re: Life Science Alliance manuscript #LSA-2021-01059

Dr. Satoshi Kanazawa
Nagoya City University Graduate School of Medical Sciences
Department of Neurodevelopmental Disorder Genetics
1 Kawasumi, Mizuho-cho, Mizuho-ku
Nagoya, Aichi 467-8601
Japan

Dear Dr. Kanazawa,

Thank you for submitting your manuscript entitled "Bimodal fibrosis in a novel mouse model of bleomycin-induced usual interstitial pneumonia" to Life Science Alliance. The manuscript was assessed by expert reviewers, whose comments are appended to this letter. We invite you to submit a revised manuscript addressing the Reviewer comments.

Thank you for this interesting contribution to Life Science Alliance. We are looking forward to receiving your revised manuscript.

Sincerely,

B. MANUSCRIPT ORGANIZATION AND FORMATTING:

Reviewer #1 (Comments to the Authors (Required)):

The overall idea of this study is interesting, the data are voluminous and of good quality, and the findings are promising. There, however, numerous, very serious concerns that need to be addressed, as outlined below.

Major concerns and suggestions

1. The manuscript describes a very large and extremely complex dataset that was generated in a vast, excessively ambitious, and poorly controlled study. There are numerous independent variables in this study: D1CC phenotype, D1BC phenotype, DBA/1J background phenotype, bleomycin, microbubbles, and sonoporation. None of these is independently controlled. For example, the BMS approach should be compared with bleomycin alone, bleomycin combined with microbubbles, and bleomycin combined with sonoporation. All of these should have been tested in each D1CC alone, D1BC alone, DAB/1J alone, D1CC on DAB/1J, D1BC on DBA/1J, and finally D1CCxD1BC on DAB/1J strains. Of course, such a design describes an entire scientific program rather than a single study. Further complicating the study, the readouts are very diverse, yet descriptive. The various readouts, while directly relevant to ILD, are not linked to each other mechanistically or logistically in this manuscript. The authors are strongly encouraged to not attempt including all of the data into this single report, because with such broad scope it comes through as unnecessarily complex yet superficial. There is enough data here for a series of articles. Perhaps this specific manuscript should be much simplified. State that you combined numerous known genetic (D1CC, D1BC, DBA/1J) and interventional (bleomycin delivery facilitated by BMS) modalities into a single model, and observed interesting, relevant results. Then exclude all molecular data (SP-D, yH2AX, S100A4, PDGFR ligands, E-cadherin, MMP7, Krt5, and similar other data) and focus solely on the descriptive disease dynamics in this model, mostly overall disease severity, histopathology, and accumulation of extracellular matrix. Once such a straightforward and clear report of the innovative model is published, follow up with your molecular, mechanistic findings in separate article(s).
2. The central notions underlying this entire report are those of UIP and NSIP. It is therefore important to define these histological patterns with utmost precision. The current definition of UIP in the introduction is far incomplete. The UIP pattern is centrally characterized by geographic heterogeneity, i.e., patchy interstitial fibrosis with alternating areas of normal lung. In IPF, the UIP pattern is pronounced mostly in subpleural, basolateral, areas of the lungs. Furthermore, in UIP, fibroblastic foci are scattered across areas of dense acellular collagen. The mentioned in the current version of the manuscript honeycombing and bronchiolization of alveoli are also present but are not by themselves solely defining of the UIP histological pattern. Substantial changes to the text and figures are necessary to provide a succinct description of the UIP, as well as NSIP, histological patterns. The authors are strongly encouraged to consult authoritative sources and briefly but completely and definitively characterize the UIP and NSIP histological patterns. Then, the claims of NSIP and UIP should be clearly supported by detailed histopathological data in the proposed model. It is important to show histological changes at low and high magnifications, as is customary in supporting both UIP and NSIP claims. Specific features of the UIP and NSIP patterns should be indicated with arrows and arrowheads in the corresponding figure panels.
3. There are numerous missed opportunities to show important experimental findings. What was the dynamics of the total body weight? Of total and differential cell counts in the BAL? Lung wet / dry weight? Total lung protein or total BAL protein? Total lung collagen measured by hydroxyproline?
4. The critical data in Figs. 3A and S4 are questionable. Trichrome staining, by definition, results in a three-colored pattern, with dark purple nuclei, red muscle, and blue-to-green connective tissues. The shown images appear to be colored only in blue, thus appearing monochrome, not trichrome.
5. For Fig. 1C - E and similar panels, show the actual western blots in supplemental figures.
6. The Discussion section is excessively voluminous. This is not a review article. Remove all broadly general statements from the Discussion, e.g., "IPF is a chronic progressive disease," and similar declarations. Furthermore, all relevant background should be included in a succinct form in the Introduction. In the Discussion, only discuss the actual specific findings within the current scientific framework of the relevant literature, arriving to an overall conclusion

Reviewer #2 (Comments to the Authors (Required)):

- 1) In this paper, the authors have characterised a transgenic model for usual interstitial pneumonia (UIP). In addition, they have used an unique delivery method to increase the disease modifying effects of bleomycin following a single intra-tracheal instillation. By using conventional histological, immunohistochemical and Western blot techniques, the authors have shown these bleomycin-treated transgenic mice exhibit bimodal lung fibrosis. I believe this paper advances the field of respiratory

medicine as it describes a novel transgenic model that overcomes some of the limitations of current animal models by displaying a disease phenotype more similar to the human condition. However, there are a couple of experimental details that need to be addressed to fully justify the conclusions drawn in this paper.

Specific points:

2a) The authors first showed that serum levels of SP-D are elevated after bleomycin treatment of their transgenic mice by ELISA (Figure 1B) as a surrogate marker for IP induction. How was the serum SP-D cut-off value determined in this figure? Is this the limit of detection by the ELISA?

2b) The authors have then shown that BMS induces DNA damage to alveolar and bronchial epithelial cells and not to fibroblasts (Figure 2) and there is more fibrosis (Figure 3) from histology/IF images. In general, I question the robustness of the way the authors have selected sections for counting γ H2AX+ cells (Figure 2) and determining the degree of fibrosis by the Ashcroft score (Figure 3). In the conventional single-dose bleomycin i.t instillation models, the pattern of fibrosis tended to occur more in the top part of the lung than the base. By selecting 5 random images of cross-sectional cut lung to count γ H2AX or score fibrosis, the authors may be inadvertently underestimating the cell count or fibrosis by randomly including lower lung sections where the bleomycin may have not reached to induce damage. Longitudinal cut sections would be better to represent the whole lung from top to bottom. In addition, if the sonoporation process improves the distribution of bleomycin, longitudinal cut sections of the lung should be included to show even distribution of fibrosis.

2c) In the "BMS induces bimodal fibrosis characterised by acute inflammation and chronic honeycombing" results section (page 18-19), the authors state "notably, severe fibrosis with honeycombing structure, but not marked...", however this is not linked to any histology data. Please indicate areas of honeycombing in figure 3A and refer in text.

2d) In the "BMS induces bimodal fibrosis characterised by acute inflammation and chronic honeycombing" results section (page 20), it is stated TGF-beta was increased at wk14 by Western blot (Fig. 3J). However there is no significance shown on the graph to justify there is more TGF-beta at wk14 than wk0. Please correct statement in text.

2e) The authors then provide robust evidence to show that non-proliferating fibroblasts are found in the fibrotic lesions (Figure 4) and epithelial cells after BMS induction have an invasive phenotype (Figure 5).

2f) Lastly, the authors examine vascularization in their animal model using IHC, Western blot and micro-CT to strongly support there is no new blood vessel formation after BMS induction (Figure 6). However, they state on page 23 "...Acute BMS-induced decreased in VEGF and CD31 expression...". From Fig. 6D, there is no significance shown on this graph to justify levels of VEGF are different at any timepoint compared to 0w. Please correct this statement in text. In addition figure 6G and 6H are the wrong way round, please correct the figure & legend accordingly.

Minor points:

3a) On page 5, Line 2 - "...intratracheal (i.t.) instillation..." intratracheal should be hyphenated.

3b) On page 5, Line 11 - should be short time-frame (hyphen is in the wrong place)

3c) On page 5 and page 6, please correct reference for B et al., 2013. (this should be Moore BB et al., 2013).

3d) In the "Mice & BMS administration" methods section (page 8, line 12), the authors state the selected bleomycin dose was determined from pilot dose-response experiments. However, this data is missing. Please include this in the supplemental data.

3e) In the "Western blot" methods section (page 9), some of the primary antibodies chosen are made in mice or goats. What secondary antibodies were used to develop these and if an anti-mouse HRP secondary antibody was used, how was non-specific background controlled for?

3f) In the "micro-CT" section of the methods (page 14), please clarify how the density of blood vessels was calculated.

3g) On page 17, line 7 - please include hyphen between "bleomycin" and "induced".

3h) On page 24, line 1 and page 36, line 11 - "ex vivo" should be in italics.

3i) On legend for Figure 3 (page 33) - please state what colours are shown by the Masson's Trichrome stain.

September 26, 2021

We thank the reviewers for their careful and thoughtful criticism. We have addressed all the reviewers' points and have included new data in the revised manuscript.

Reviewer 1 (R1) comments:

Comment: The overall idea of this study is interesting, the data are voluminous and of good quality, and the findings are promising. There, however, numerous, very serious concerns that need to be addressed, as outlined below.

Response to R1 comment: We thank the reviewers for their interest in our study, which has established a novel iUIP mouse model. We have responded to the reviewer's comments to further improve the quality of our data in detail below.

R1 Major concerns and suggestions 1a:

The manuscript describes a very large and extremely complex dataset that was generated in a vast, excessively ambitious, and poorly controlled study. There are numerous independent variables in this study: D1CC phenotype, D1BC phenotype, DBA/1J background phenotype, bleomycin, microbubbles, and sonoporation. None of these is independently controlled. For example, the BMS approach should be compared with bleomycin alone, bleomycin combined with microbubbles, and bleomycin combined with sonoporation. All of these should have been tested in each D1CC alone, D1BC alone, DAB/1J alone, D1CC on DAB/1J, D1BC on DBA/1J, and finally

D1CC×D1BC on DAB/1J strains. Of course, such a design describes an entire scientific program rather than a single study.

Response to R1 comment 1a: We thank the reviewer for these insightful suggestions.

We chose to perform the majority of the studies using D1CC×D1BC mice, on the basis of our previous publications, which showed susceptibility to development of rheumatoid arthritis-associated interstitial pneumonia, RA-ILD in D1CC mice (Kanazawa et al PNAS 2016, Terasaki et al J Cell Mol Med 2019).

As acknowledged, the full range of suggested comparisons in multiple strains would be beyond the scope of this study. However, we showed that BMS administration induced IP (SP-D at 2 weeks) in IP-susceptible D1CC×D1BC mice, but was not sufficient in D1CC, D1BC or DBA/1J mice (Figure S7). We have also recently published a new data, which shows more severe RA-ILD in D1CC×D1BC mice (Miura et al ERJ open research 2021 in press)

Having selected D1CC×D1BC mice as the IP-susceptible strain, our design has controlled for many of the independent variables identified by the reviewer. In the D1CC×D1BC mice, we have provided results for the time course of SP-D following several doses of bleomycin in BMS (new Figure S1) as well as serum SP-D at 2 weeks, comparing (1) sonoporation alone (2) microbubbles alone (3) sonoporation and microbubbles (4-6) of BMS with three lower doses of bleomycin (0.008-0.2 mg/kg) (Fig S2). The similar levels of basal SP-D in all groups demonstrate the lack of effect of sonoporation and/or microbubbles, while the dose-ranging justifies the use of the selected dose of 1.25 mg/kg bleomycin in BMS.

Critically, we also showed that the increase in SP-D over 14 weeks with 1.25 mg/kg bleomycin alone was much lower than the same dose administered as BMS (Figure 1B) and that even a 3-fold higher dose of bleomycin alone (without microbubbles or sonoporation) did not increase TUNEL staining in the D1CC×D1BC strain (Figure S3).

R1 Major concerns and suggestions 1b:

Further complicating the study, the readouts are very diverse, yet descriptive. The various readouts, while directly relevant to ILD, are not linked to each other mechanistically or logistically in this manuscript. The authors are strongly encouraged to not attempt including all of the data into this single report, because with such broad scope it comes through as unnecessarily complex yet superficial. There is enough data here for a series of articles. Perhaps this specific manuscript should be much simplified.

State that you combined numerous known genetic (D1CC, D1BC, DBA/1J) and interventional (bleomycin delivery facilitated by BMS) modalities into a single model, and observed interesting, relevant results. Then exclude all molecular data (SP-D, H2AX, S100A4, PDGFR ligands, E-cadherin, MMP7, Krt5, and similar other data) and focus solely on the descriptive disease dynamics in this model, mostly overall disease severity, histopathology, and accumulation of extracellular matrix. Once such a straightforward and clear report of the innovative model is published, follow up with your molecular, mechanistic findings in separate article(s).

Response to R1 Major concerns and suggestions 1b:

We thank the reviewer for acknowledgment of our “innovative model”. We have presented the overall characteristics of the iUIP model, rather than limit our findings to descriptive dynamics, as we believe that the inclusion of consolidated data showing detailed histochemical, biochemical and computed micro-CT analyses is necessary to help the reader interpret some of the clinical data.

For example, the differences in the time course of SP-D levels in serum and lung (Fig. 1B and 1E) suggest that while serum SP-D is useful biomarker to determine the early onset of IP, it does not relate to the progress of IP as defined by the cellular markers in our other data. Furthermore, our comprehensive analysis allowed us to propose the interesting conclusion that the development of IPF may be represented by a continuous progression from NSIP to UIP phases. Additional insights arise from the observations of rheumatoid arthritis-associated interstitial pneumonia, RA-ILD (now published as Miura et al ERJ open research 2021 in press) as well as IPF in D1CC×D1BC mice presented here. In both RA and UIP models, invasive epithelial cells expressing EMT-related molecules such as γ 2pf and MMP7 were observed along with cells close to basal cells (Fig. 5). Because abnormalities in these epithelial cells may affect fibrosis, we are working to further characterize them.

R1 Comment 2a: The central notions underlying this entire report are those of UIP and NSIP. It is therefore important to define these histological patterns with utmost precision. The current definition of UIP in the introduction is far incomplete. The UIP pattern is centrally characterized by geographic heterogeneity, i.e., patchy interstitial fibrosis with alternating areas of normal lung. In IPF, the UIP pattern is pronounced mostly in subpleural, basolateral, areas of the lungs. Furthermore, in UIP, fibroblastic foci are scattered across areas of dense acellular collagen. The mentioned in the current version of the manuscript honeycombing and bronchiolization of alveoli are also present but are not by themselves solely defining of the UIP histological pattern. Substantial

changes to the text and figures are necessary to provide a succinct description of the UIP, as well as NSIP, histological patterns. The authors are strongly encouraged to consult authoritative sources and briefly but completely and definitively characterize the UIP and NSIP histological patterns. Then, the claims of NSIP and UIP should be clearly supported by detailed histopathological data in the proposed model.

Response to R1 comment 2a: We thank the reviewer for this additional information. Our descriptions of NSIP and UIP in iUIP model have been revised in the introduction. We now briefly introduce histopathologic definitions of NSIP as well as UIP with appropriate references.

The manuscript has been revised with additional text as follows:

Introduction (page 5 line 4- 8): Nonspecific interstitial pneumonia, NSIP as well as UIP, shows various clinical, pathologic, and radiologic manifestations and a typical histopathological pattern. Most cases of NSIP show fibrosis with chronic inflammatory cells consisting of lymphocytes and other cells, but there are some exceptions (Tabaj et al 2015, Travis et al 2008).

Introduction (page 4 lines 10-15): The UIP pattern is characterized by geographic heterogeneity, with patchy interstitial fibrosis alternating with normal parenchyma. In IPF, the UIP pattern is found mainly in subpleural, basolateral areas of the lungs, with scattered foci of fibroblasts in areas of dense acellular collagen (Smith et al 2013). Most of honeycomb cysts were surrounded by hyperplastic epithelium, and associated with traction bronchiolectasia (Hashisako & Fukuoka 2015).

R1 Comment 2b: It is important to show histological changes at low and high magnifications, as is customary in supporting both UIP and NSIP claims. Specific features of the UIP and NSIP patterns should be indicated with arrows and arrowheads in the corresponding figure panels.

Response to R1 comment 2b: Thank you for this suggestion which we have addressed with the provision of additional high-power images and annotations to figure panels.

In terms of data for NSIP phase, at this initial stage, cell death by bleomycin DNA double-strand breaks and subsequent infiltration of lymphoid cells were observed. Since infiltrated lymphocytes were observed at one week after BMS induction, a new magnified image of day 7 was added to Figure S4A. Details of infiltrated lymphoid cells were analyzed in immunohistochemical data in Figure S6. At NSIP stage, more macrophages and T cells were observed. The new images for each stained data were added as Figures S6E and S6F.

For the data of UIP phase, especially the data showing honeycombing with less infiltrated lymphocytes, we were technically unable to perform 3D imaging of the lung by micro-CT using live animals. However, histological analysis clearly shows that the honeycombs are constructed along with hyperplasia of the epithelium surrounding the basolateral area. We observed a patchy honeycombing only during chronic phase, 10 weeks after BMS induction, which is shown in S5. This characteristic structure was also reflected in Ashcroft score, with most of the lung samples at weeks 10 and 14 showing high score in Figure 3C.

Histopathological images of NSIP (week 2) and UIP (week 14) stained by Masson's trichrome were replaced by magnified images in Figure 3A. Each image was represented severe inflammation at NSIP phase and a typical honeycombing at UIP. As noted in the comparison of the histological features among DBA/1J, D1CC, D1BC, D1CC×D1BC mice in Figure S7, we only observed honeycombing in D1CC×D1BC mouse. In all figures including a new Masson-staining data, which is replaced in Fig. 3A, honeycomb structures were marked as arrows. In addition, we point out subpleural hyperplasia, for example, hyperplastic area is widely extended at week 14 in Figure 3I. Thus, this region is indicated by an asterisk.

The manuscript has been revised with additional text and figures as follows:

Figure 3A (page 33 lines 17- page 34 line 2) and associated figure legends:

Representative histopathological micro-graphs of Masson's trichrome stained sections from BMS-induced D1CC×D1BC mice. In acute phase at week 2, many infiltrating cells were observed. In the chronic phase at week 14, honeycombing was indicated by yellow arrows.

Figure 3I and associated figure legend (page 34 lines 13): Asterisks indicates subpleural hyperplasia.

Supplemental Figure S4 and figure legend: Lung sections at 0, 6, and 24 hours, and 3 and 7 days with magnified image.

Supplemental Figure S5 and figure legends: Arrows indicate honeycombing at weeks 10 and 14.

Supplemental Figures S6E and S6F, and figure legends: (E) Each image represents immunohistochemical staining of (E) macrophages (F4/80, green), T cells (CD3, red), (F) neutrophils (PAD4, green), and B cells (CD45R, red). Scale bars = 100 μm.

Supplemental figure S7 and figure legends: Representative histopathological micro-graphs of Masson's trichrome stained sections from BMS-induced D1CC×D1BC mice. In acute phase at week 2, many infiltrating cells were observed. Arrows indicate honeycombing in the chronic phase at week 14.

R1 comment 3: There are numerous missed opportunities to show important experimental findings. What was the dynamics of the total body weight? Of total and differential cell counts in the BAL? Lung wet / dry weight? Total lung protein or total BAL protein? Total lung collagen measured by hydroxyproline?

Response to R1 comment 3: We acknowledge the reviewer's comments. We now present the dynamics of total body weight and whole lung weight at each stage, shown as additional panels in Figure 1 (new Fig 1C and 1D). On average, body weight decreased by 5~10% in D1CC×D1BC mice during the acute phase following BMS, and slowly recovered until week 14. The total lung weight (Fig 1C) showed a bimodal pattern similar to that of the fibrosis data, shown in Figure 3B, C, E, F, G, and H.

We did not analyze BAL samples from each stage in this paper. However, histopathological analysis in Figure S6 demonstrated localization and distribution of lymphoid cells in the lung at each stage. Thus, this data gives an information similar to flowcytometric analysis using BAL samples.

The manuscript has been revised with additional text or figures as follows:

Results (page 17 lines 14-16): Body weight decreased by approximately 5~10% during the acute phase following BMS administration in D1CC×D1BC mice, and slowly recovered until week 14 (Fig 1C). Total lung weights showed a bimodal pattern consistent with all measures of lung fibrosis (Fig 1D).

Figure 1 and figure legends (page 32 lines 9-12): (C and D) Body weight variability ratio (C) and total lung weight (D) after BMS administration in D1CC×D1BC mice. Data are mean ± S.E. from nine mice per group. Asterisks show * $P < 0.05$ and **** $P < 0.0001$. The alphabetical indexes in the text and Figure legend have been changed accordingly.

R1 comments 4: The critical data in Figs. 3A and S4 are questionable. Trichrome staining, by definition, results in a three-colored pattern, with dark purple nuclei, red muscle, and blue-to-green connective tissues. The shown images appear to be colored only in blue, thus appearing monochrome, not trichrome.

Response to R1 comment 4: We acknowledge the reviewer's comments. Masson's trichrome reagents from Muto pure chemicals (Tokyo) were used for staining. The staining method followed the manufacture's protocol. The optimal staining condition for aniline blue to stain collagen fibers, which is used for evaluation of "fibrosis (%)" and Ashcroft score in Figure 3B and 3C, was set as 20 minutes (10-30 minutes in the manufacturer's protocol). This resulted in a slightly stronger staining of blue, but

microscopic evaluation confirmed that the different stains could be detected for selective analysis of fibrosis by Masson's trichrome staining.

The manuscript has been revised and the following figures have been replaced: Fibrosis ratio (page 13 line 18-page14 line2): Images showing the area of fibrosis represented in blue (ECM-deposition) by Masson's trichrome staining (20 minutes for aniline blue staining) were captured by BZ-X analyzer (Keyence) and analyzed by ImageJ, Fiji.

Figure 3A and figure legend (page 33 lines 17- page 34 line 2): Masson's trichrome stained images in Figure 3A has been replaced by a new magnified image, as described above.

R1 comments 5: For Fig. 1C - E and similar panels, show the actual western blots in supplemental figures.

Response to R1 comment 5: We acknowledge the reviewer's comments. All the actual WB data were added in Figure S10 as new supplementary data.

The manuscript has been revised with additional figure as follows:

Supplemental figure S10 and figure legend: Protein expression in whole lung extracts determined by Western blot and visualized by ECL chemiluminescence system. The relative signal intensity for each protein was determined using tubulin- α as a loading control (shown for one blot only, bottom). Antibody details are provided in Materials and Methods. Grouped data from three mice at weeks 0, 2, 8, and 14 is shown in Figure 1E to 1G (SP-A, SP-C, SP-D), Figure S8 (α SMA), Figure 3G (Type II collagen), Figure 3J (TGF- β), Figure 4 (Vimentin, PDGFR α , PDGFR β , PDGF α , PDGF β), Figure 5 (E-cadherin, uteroglobin), and Figure 6 (CD31, VEGFA, PEDF). Arrows indicate each protein.

R1 comments 6: The Discussion section is excessively voluminous. This is not a review article. Remove all broadly general statements from the Discussion, e.g., "IPF is a chronic progressive disease," and similar declarations. Furthermore, all relevant background should be included in a succinct form in the Introduction. In the Discussion, only discuss the actual specific findings within the current scientific framework of the relevant literature, arriving to an overall conclusion

Response to R1 comment 6: We revised introduction and discussion according the reviewer's comments. Repeated sentences with citation were removed from the discussion. "IPF is a chronic, progressive interstitial lung disease that can occur as a

result of prolonged IP.” and subsequent sentences revised into Introduction from Discussion.

The manuscript has been revised as follows:

Discussion: “using higher or repeated doses of intra-tracheal bleomycin (Sato et al 2017, Tashiro et al 2017).” was deleted.

Introduction (page 4 lines 3-7): Idiopathic pulmonary fibrosis (IPF) is classified as one of the most serious chronic and progressive IPs leading to death due to the loss of pulmonary function (Antoniou et al 2014). Fibrotic foci in IPF are often covered by a cuboidal lining epithelium, attributed to squamous metaplasia of epithelial cells (Batra et al 2018, Hashisako & Fukuoka 2015, Hogan et al 2014).

Discussion: “, where chronic IP is not commonly established without repeated bleomycin dosing (Degryse et al 2010, Tashiro et al 2017).” was deleted.

Reviewer 1 (R1) comments:

Reviewer 2 (R2) comments 1: In this paper, the authors have characterized a transgenic model for usual interstitial pneumonia (UIP). In addition, they have used an unique delivery method to increase the disease modifying effects of bleomycin following a single intra-tracheal instillation. By using conventional histological, immunohistochemical and Western blot techniques, the authors have shown these bleomycin-treated transgenic mice exhibit bimodal lung fibrosis. I believe this paper advances the field of respiratory medicine as it describes a novel transgenic model that overcomes some of the limitations of current animal models by displaying a disease phenotype more similar to the human condition. However, there are a couple of experimental details that need to be addressed to fully justify the conclusions drawn in this paper.

Response to R2 comment: We thank the reviewers for their interest in our study, which has established a novel iUIP mouse model. We have responded to the reviewer’s comments in detail below.

Specific points:

R2 comments 2a: The authors first showed that serum levels of SP-D are elevated after bleomycin treatment of their transgenic mice by ELISA (Figure 1B) as a surrogate marker for IP induction. How was the serum SP-D cut-off value determined in this figure? Is this the limit of detection by the ELISA?

Response to R2 comment 2a: The serum concentration of SP-D was measured in more than 20 BMS-untreated D1CC×D1BC mice, and the mean serum concentration of SP-D plus two times the standard deviation was used as the cutoff value of SP-D, which was 53.9ng/ml. The criteria for serum SP-D cut of value are described in Methods (P10). The manuscript has been revised with additional figure as follows:

Methods (page 10 lines 1-4): The serum concentration of SP-D was measured in more than 20 BMS-untreated D1CC×D1BC mice, and the mean serum concentration of SP-D plus two times the standard deviation was used as the cutoff value of SP-D, which was 53.9ng/ml.

R2 comments 2b: The authors have then shown that BMS induces DNA damage to alveolar and bronchial epithelial cells and not to fibroblasts (Figure 2) and there is more fibrosis (Figure 3) from histology/IF images. In general, I question the robustness of the way the authors have selected sections for counting γ H2AX+ cells (Figure 2) and determining the degree of fibrosis by the Ashcroft score (Figure 3). In the conventional single-dose bleomycin i.t instillation models, the pattern of fibrosis tended to occur more in the top part of the lung than the base. By selecting 5 random images of cross-sectional cut lung to count γ H2AX or score fibrosis, the authors may be inadvertently underestimating the cell count or fibrosis by randomly including lower lung sections where the bleomycin may have not reached to induce damage. Longitudinal cut sections would be better to represent the whole lung from top to bottom. In addition, if the sonoporation process improves the distribution of bleomycin, longitudinal cut sections of the lung should be included to show even distribution of fibrosis.

Response to R2 comment 2b: We acknowledge the reviewer's comments. To estimate the distribution of bleomycin delivery to the lung, a preliminary test was performed using a vehicle solution containing a coloring agent by a spray nebulizer (Natsume, Japan). The dye was shown to be spread evenly throughout the lung with a single shot if the insertion of the nebulizer was correct. Also, to spread bleomycin evenly in the lung, mice remained for at least 3 minutes under anesthesia after delivery before sonoporation. Under these conditions, there was no tendency for the fibrosis to be biased toward the upper part of the lung. Thus, the manuscript has been revised as follows.

Methods (page 9 lines 6-13): Bleomycin (0.512 mg/mL in normal saline, Nippon Kayaku) was mixed with an equal amount of microbubbles (Ultrasound Contrast Agent SV-25, NepaGene) and administered via the intra-tracheal (i.t.) route by a spray nebulizer (40 μ l/mouse, 1.28 mg/kg, Natsume) prior to sonoporation on the chest by 1.0

W/cm² for 1 minute (Sonitron GTS Sonoporation System, NepaGene). Induction of IP in response to BMS was monitored by measurement of serum SP-D levels. **Preliminary tests confirmed that this resulted in uniform distribution of bleomycin in the lung, and therefore reduced tendency to bias the location of lesion sites to the upper lung.**

R2 comments 2c: In the "BMS induces bimodal fibrosis characterized by acute inflammation and chronic honeycombing" results section (page 18-19), the authors state "notably, severe fibrosis with honeycombing structure, but not marked...", however this is not linked to any histology data. Please indicate areas of honeycombing in figure 3A and refer in text.

Response to R2 comment 2c: We acknowledge the reviewer's comments. As in #1 reviewers' comment 2, we revised the manuscript by revised text and figures about NSIP and UIP including Figure 3A. Please refer to the above response.

R2 comments 2d: In the "BMS induces bimodal fibrosis characterized by acute inflammation and chronic honeycombing" results section (page 20), it is stated TGF-beta was increased at wk14 by Western blot (Fig. 3J). However, there is no significance shown on the graph to justify there is more TGF-beta at wk14 than wk0. Please correct statement in text.

Response to R2 comment 2d: We acknowledge the reviewer's comments. Since the WB data of TGF-β does not show significant P value, it was revised as follows.

Results (page 21 lines 6-8): **It was not significant, however, this tended to increase only at week 14, but not at weeks 2 and 8 (Fig 3J).**

R2 comments 2e: The authors then provide robust evidence to show that non-proliferating fibroblasts are found in the fibrotic lesions (Figure 4) and epithelial cells after BMS induction have an invasive phenotype (Figure 5).

Response to R2 comment 2e: We thank the reviewer for this comment on our robust data.

R2 comments 2f: Lastly, the authors examine vascularization in their animal model using IHC, Western blot and micro-CT to strongly support there is no new blood vessel formation after BMS induction (Figure 6). However, they state on page 23 "...Acute BMS-induced decreased in VEGF and CD31 expression...". From Fig. 6D, there is no significance shown on this graph to justify levels of VEGF are different at any timepoint compared to 0w. Please correct this statement in text. In addition, figure 6G and 6H are

the wrong way round, please correct the figure & legend accordingly.

Response to R2 comment 2f: We acknowledge the reviewer's comments. In this sentence, we would like to describe the expression of CD31 and VEGF was not restored at any phases, therefore, we revised the manuscript as follows.

Results (page 24 lines 14-16): **Acute BMS-induced decreases in CD31 and VEGF expression were not restored in later phases after BMS administration (Figs 6C, 6D).**

Minor points:

R2 comments 3a: On page 5, Line 2 - "...intratracheal (i.t.) instillation..." intratracheal should be hyphenated.

Response to R2 minor comment 3a: Thank you, this has now been corrected:

Introduction (page 5 line 14): "intratracheal (i.t.) instillation" in the text has been changed to "**intra-tracheal** (i.t.) instillation".

R2 comments 3b: On page 5, Line 11 - should be short time-frame (hyphen is in the wrong place)

Response to R2 minor comment 3b: This has now been corrected as follows:

Introduction (page 6 line 5): "short-time frame" in the text has been changed to "**short time-frame**".

R2 comments 3c: On page 5 and page 6, please correct reference for B et al., 2013. (this should be Moore BB et al., 2013).

Response to R2 minor comment 3c: This has now been corrected adding the author's name to the references.

R2 comments 3d: In the "Mice & BMS administration" methods section (page 8, line 12), the authors state the selected bleomycin dose was determine from pilot dose-response experiments. However, this data is missing. Please include this in the supplemental data.

Response to R2 minor comment 3d: We acknowledge the reviewer's comments. Three different concentrations of bleomycin (0.96, 1.28, 1.6 mg/kg) were examined in D1CC×D1BC. We revised a data about the incidence of interstitial pneumonia at different concentrations of bleomycin in the BMS method as a supplemental figure S1. D1CC×D1BC mice treated with either 1.28 or 1.6 mg/kg-treated demonstrated higher incidence of interstitial pneumonia than the lower concentration of bleomycin (0.96mg/

kg). Approximately less than 10 % weight loss at one to two weeks after BMS administration was observed in both of 1.28 and 1.6mg/ kg-treated mice. The results, including the concentration of serum SP-D, are shown in a new supplemental table S1. However, there were no significant differences between them in terms of body weight and serum SP-D. Therefore, the concentration of bleomycin was set at 1.28 mg/ kg, which is sufficient concentration for IP induction.

The manuscript has been revised as follows:

Results (page 17 line 9): The selected bleomycin dose was the mid-range of dose-ranging pilot studies using 0.96-1.6 mg/kg body weight with less body weight loss (Fig S1 and Table S1).

Supplemental Figure S1: Figure S1 Incidence of IP with different concentration of bleomycin in BMS method

The incidence of IP was determined by whether the concentration of serum SP-D exceeded the cut-off value (53.9 ng/ml). Bleomycin was administered intratracheally at 0.96 (closed triangle), 1.28 (closed square), and 1.60 mg/kg (closed circle), respectively. Results are represented as means \pm S.E. of ten mice per group.

Supplemental table S1: Supplemental Table S1: Table S1 The change of body weight and serum SP-D concentrations with different bleomycin concentrations after BMS administration

R2 comments 3e: In the "Western blot " methods section (page 9), some of the primary antibodies chosen are made in mice or goats. What secondary antibodies were used to develop these and if an anti-mouse HRP secondary antibody was used, how was non-specific background controlled for?

Response to R2 minor comment 3e: We acknowledge the reviewer's comments. Secondary antibodies to mouse and goat primary antibodies have been listed in Methods.

Methods (page 11 lines 2-5): ECLTM anti-rabbit, anti-mouse, or anti-goat IgG, horseradish peroxidase linked antibodies were used as a secondary antibody (GE Healthcare). Goat anti-mouse IgM-HRP was used as a secondary antibody for mouse anti Tubulin- α (Santa Cruz).

R2 comments 3f: In the "micro-CT" section of the methods (page 14), please clarify how the density of blood vessels was calculated.

Response to R2 minor comment 3f: We acknowledge the reviewer's comments. The sentence of "the density of blood vessels was calculated." has been revised to new

sentences. Also, in Figure 6H, we replaced the labels of the vertical and horizontal bar, respectively.

Methods (page 15 line 14-16): All micro-CT images were acquired using the whole lung with a SkyScan 1276 (Bruker). Each volume was calculated for micro vessels with a diameter of less than 100 μm .

Figure 6, figure legend (page 36 lines 14-15): (H) The total volume was calculated for each range of micro vessel diameter in the whole lung after *ex vivo* Microfil application.

Figure 6H: The labels of the vertical and horizontal bar were revised.

R2 comments 3g: On page 17, line 7 - please include hyphen between "bleomycin" and "induced".

Response to R2 minor comment 3g: This has now been corrected.

Results (page 18 line 12): "bleomycin induced" has been revised to "bleomycin-induced".

R2 comments 3h: On page 24, line 1 and page 36, line 11 - "ex vivo" should be in italics.

Response to R2 minor comment 3h: This has now been corrected (and also revised in Figure legend of Figure 6).

Results (page 25, line 3) and Figure legend of Figure 6 (page 36 line 15): "ex vivo" has been revised to "*ex vivo*".

R2 comments 3i: On legend for Figure 3 (page 33) - please state what colors are shown by the Masson's Trichrome stain.

Response to R2 minor comment 3i: We acknowledge the reviewer's comments. As mentioned at "**Response to R1 comment 4**", we focus on the staining of collagens in fibrotic regions. Therefore, blue color from Masson's trichrome staining was selectively captured and analyzed for fibrosis % in Figure 2B.

October 13, 2021

RE: Life Science Alliance Manuscript #LSA-2021-01059R

Dr. Satoshi Kanazawa
Nagoya City University Graduate School of Medical Sciences
Department of Neurodevelopmental Disorder Genetics
1 Kawasumi, Mizuho-cho, Mizuho-ku
Nagoya, Aichi 467-8601
Japan

Dear Dr. Kanazawa,

Thank you for submitting your revised manuscript entitled "Bimodal fibrosis in a novel mouse model of bleomycin-induced usual interstitial pneumonia". We would be happy to publish your paper in Life Science Alliance pending final revisions necessary to meet our formatting guidelines, as well as addressing the Reviewers' remaining concerns, including that the manuscript would benefit from additional proofreading to improve clarity.

- please upload your supplementary figures as single files as well
- please add your main, supplementary figure, and table legends to the main manuscript text after the references section
- please add the Twitter handle of your host institute/organization as well as your own or/and one of the authors in our system
- please consult our manuscript preparation guidelines <https://www.life-science-alliance.org/manuscript-prep> and make sure your manuscript sections are in the correct order
- please add callouts for Figures S4A-B; S6A-F; S7A, B; S8A, B; S10 to your main manuscript text

Figure checks:

- Figure S1 is shown as mean +/- SE, but there are no error bars shown
- please indicate sizes next to the protein blots in figure S10. Also, these blots should be displayed with their own loading controls, not just one representative loading control blot.
- In Figure 5C, the magnified sections a and b are reversed

A. FINAL FILES:

B. MANUSCRIPT ORGANIZATION AND FORMATTING:

Sincerely,

Reviewer #1 (Comments to the Authors (Required)):

The Authors have addressed the main concerns. Additional minor considerations are recommended, to be considered jointly by the Authors and the Editor:

1. The manuscript needs additional editing for English. There are multiple instances of confusing language in the current version, too numerous to be mentioned here. Some examples include the following.

The title of Figure S10, "Actual WB data," sounds confusing, and may be misconstrued as implying that some data in this manuscript are not "actual." The title should be changed to avoid the colloquial use "actual" to, e.g., "Representative western blotting images of indicated targets" or a similar title.

The statement that "NSIP as well as UIP, shows various clinical, pathologic, and radiologic manifestations and a typical histopathological pattern" is misleading on several levels. First, "pathologic" and "histopathological" are synonymous in this context, and it is not clear how one can be "various" and the other "typical." Second, despite some degree of variability, the radiologic correlates for UIP and NSIP are rather well defined, and an experienced radiologist would be able to successfully predict the histological pattern based on high-resolution chest CT imaging. Third, this sentence may appear to equate UIP and NSIP in part, which is misleading because these are distinct entities. Recommend a clearer statement, e.g., "NSIP and UIP are distinct histopathological patterns commonly observed in various interstitial pneumonias, with UIP being characteristic of IPF." In the Abstract and Introduction, IPF is not just "classified", it "is" the most serious and deadliest form of interstitial pneumonia; the use of the superfluous "classified" in this context may imply a possibility of more serious forms of interstitial pneumonias, and such do not exist.

In line 9 on page 6, "Assemble ... are usually assessed" is unclear.

These are by far not all examples of confusing statements. Additional editing of the manuscript throughout the entire text is recommended to add clarity.

2. Consider selecting different trichrome sections for fig. 3A. That panel is still different (too much blue) from the commonly observed appearance of trichrome staining (which can be easily seen by performing Google image search for << trichrome lung bleomycin >>. The current fig. S5, while still imperfect, looks closer to how a trichrome staining usually appears; why not use one of those or similar sections for panel 3A?

Reviewer #2 (Comments to the Authors (Required)):

Comments to the authors:

I have re-read manuscript LSA-2021-01059R and I am satisfied that the authors have addressed my comments (R2 specific points, points 2a-2f, 3a-3i). I thank the authors for clarifying my queries and for updating the manuscript accordingly. I believe this manuscript is now suitable for publication in this journal.

Referee cross comments:

I agree with Reviewer 1 that the manuscript would benefit from additional proof-reading services to improve the clarity of the writing style.

October 21, 2021

RE: Life Science Alliance Manuscript #LSA-2021-01059RR

Dr. Satoshi Kanazawa
Nagoya City University Graduate School of Medical Sciences
Department of Neurodevelopmental Disorder Genetics
1 Kawasumi, Mizuho-cho, Mizuho-ku
Nagoya, Aichi 467-8601
Japan

Dear Dr. Kanazawa,

Thank you for submitting your Research Article entitled "Bimodal fibrosis in a novel mouse model of bleomycin-induced usual interstitial pneumonia". It is a pleasure to let you know that your manuscript is now accepted for publication in Life Science Alliance. Congratulations on this interesting work.

DISTRIBUTION OF MATERIALS:

Again, congratulations on a very nice paper. I hope you found the review process to be constructive and are pleased with how the manuscript was handled editorially. We look forward to future exciting submissions from your lab.

Sincerely,
